



# The influence of weather-driven processes on tropospheric ozone

Tamara Emmerichs[1], Bruno Franco[2], Catherine Wespes[2], Vinod Kumar[3], Andrea Pozzer[4], Simon Rosanka[1], and Domenico Taraborrelli[1]

[1]Institute of Energy and Climate Research, IEK-8: Troposphere, Forschungszentrum Jülich, Jülich, Germany
[2]Université libre de Bruxelles (ULB), Spectroscopy, Quantum Chemistry and Atmospheric Remote Sensing (SQUARES), Brussels 1050, Belgium
[3]Satellite Remote Sensing Group, Max Planck Institute for Chemistry, Mainz, Germany
[4]Atmospheric Chemistry Department, Max Planck Institute for Chemistry, Mainz, Germany

**Correspondence:** Tamara Emmerichs (t.emmerichs@fz-juelich.de) and Domenico Taraborrelli (d.taraborrelli@fz-juelich.de)

**Abstract.** Near-surface ozone is an harmful air pollutant, which is determined to a considerable extent by weather-controlled processes, and may be significantly impacted by water vapour forming complexes with peroxy radicals. The role of water in the reaction of $HO_2$ radical with nitrogen oxides is known from the literature, and in current models the water complex is considered by assuming a linear dependence on water concentrations. In fact, recent experimental evidence has been pub-

lished, showing the significant role of water on the kinetics of one of the most important reaction for ozone chemistry, namely $NO_2 + OH$. Here, the available kinetic data for the $HO_x + NO_x$ reactions have been included in the atmospheric chemistry model ECHAM5/MESSy (EMAC) to test its global significance. Among the modified kinetics, the newly added $HNO_3$ channel from $HO_2 + NO$, dominates, significantly reducing $NO_2$. A major removal process of near-surface ozone is dry deposition accounting for 20 % of the total tropospheric ozone loss mostly occurring over vegetation. However, parameterizations for

modelling dry deposition represent a major source of uncertainty for tropospheric ozone simulations. This potentially belongs to the reasons why global models, such as EMAC used here, overestimate ozone with respect to observations. In fact, the employed parameterization is hardly sensitive to local meteorological conditions (e.g., humidity) and lacks non-stomatal deposition. In this study, a dry deposition scheme including these features have been used in EMAC, affecting not only the deposition of ozone but of its precursors, resulting in lower chemical production of ozone. Additionally, we improved the

emissions of isoprene and nitrous acid (HONO). Namely, for isoprene emissions we have accounted for the impact of drought stress which confers a higher model sensitivity to meteorology leading to reduced annual emissions down to 32 %. For HONO, we have implemented soil emissions, which depend on soil moisture and thus on precipitation. We estimate for the first time a global source strength of $7\,Tg(N)\,a^{-1}$. Furthermore, the usage of a parameterization for the production of lightning $NO_x$ that depends on cloud top height contributes to a more realistic representation of $NO_2$ columns over remote oceans with respect to

the satellite measurements of the Ozone Monitoring Instrument (OMI). The combination of all the model modifications reduces the simulated global ozone burden by $\approx 20$ % to $337\,Tg$, which is in better agreement with recent estimates. By comparing simulation results with measurements from the Infrared Atmospheric Sounding Interferometer (IASI) and the Tropospheric Ozone Assessment Report (TOAR) databases (of 2009) we demonstrate an overall reduction of the ozone bias by a factor of 2.





## 1 Introduction

The investigation of tropospheric ozone ($O_3$) plays a prominent role in atmospheric research. In fact, $O_3$ is an air pollutant harmful for humans and plants, has a significant radiative forcing, and is an important oxidant in the troposphere (Gaudel et al., 2018; Fleming et al., 2018). Its levels are strongly affected by radical reactions and surface-atmosphere exchanges which in turn are modulated, directly and indirectly, by weather (Fiore et al., 2012; Jacob and Winner, 2009; Sadiq et al., 2017; Fu and Tian, 2019). Understanding the impact of weather on ozone, and air quality in general, is thus important also in view of weather extremes. In fact, the frequency and intensity of heat waves and droughts are projected to increase due to climate change (Hou and Wu, 2016). Global chemistry-climate models (CCM) overestimate tropospheric ozone by about 20 % in the Northern Hemisphere with respect to observations (Young et al., 2018, 2013). The global atmospheric chemistry model ECHAM/MESSy (EMAC; Jöckel et al., 2010) is no exception to this since Jöckel et al. (2016) reported that EMAC tends to overpredict tropospheric $O_3$ columns by up to 15 DU[1] (see their Fig. 29) when compared to satellite retrievals. Recently, by using the Jülich Aqueous-phase Mechanism of Organic Chemistry (JAMOC; Rosanka et al., 2021a) in EMAC, Rosanka et al. (2021b) identified that the simplified representation of in-cloud organic chemistry may account for up to 20 % of this model bias. In addition, Rosanka et al. (2020) reported that a comprehensive representation of this process is important for estimating the impact of intense peat fires on tropospheric $O_3$. Still, an ongoing model development and assessment with a focus tropospheric ozone is needed to further reduce the high model bias. The meteorological dependence of processes driving ozone has been well reported in the literature (Kavassalis and Murphy, 2017, and references therein). The ozone levels near the surface are determined by chemical production and loss reactions involving nitrogen oxides ($NO_x=NO+NO_2$) and volatile organic compounds (VOCs) (Monks, 2005). Radical reactions are known to be affected in the presence of water vapour. Water vapour is the third most abundant species in the troposphere with the largest abundance at the lowest levels. The ability to form stable complexes with atmospheric radicals is acknowledged which modifies the kinetics of the $HO_x$ ($HO_x=OH+HO_2$) and $NO_x$ reactions which are key to photo-chemical ozone production. However, the relevant kinetics is not entirely known and included only partially and in a simplified way in models (Buszek et al., 2011). Dry deposition represents an important sink accounting for 20 % of the total $O_3$ loss in the troposphere (Young et al., 2018) which is the highest over vegetation (Hardacre et al., 2015). However, the representation of dry deposition in models is uncertain due to the limited amount of observations and the dependence on input data (Val Martin et al., 2014; Wong et al., 2019; Wesely and Hicks, 2000). A potential uncertainty source is the parameterization of surface resistance as identified in a multi-model evaluation by Hardacre et al. (2015). Schwede et al. (2011) points to soil and cuticular (wax coverage of leaves) uptake. Also, the dependence on meteorology plays a major role for the prediction of dry deposition and its implementation becomes especially desirable in the light of a changing climate (Andersson and Engardt, 2010; Wong et al., 2019). A further important process affecting the chemistry and fate of tropospheric ozone is the biogenic emission of isoprene ($C_5H_8$), the most important volatile organic compound (BVOC) emitted from plants

[1] Dobson unit





(Guenther et al., 1995). Isoprene oxidation contributes to ozone production in urban and sub-urban (VOC-limited) regions and impacts the atmospheric levels of the hydroxyl radical (OH) determining the oxidation capacity of the troposphere (Monks, 2005; Pusede et al., 2015). Global estimates of isoprene emissions cover a wide range and are highly uncertain across models since the estimates depend on the spatial resolution, the used parameterizations and vegetation data (Guenther et al., 2006). The

relevant dependencies for the parameterization of isoprene emissions are known (Sindelarova et al., 2014; Müller et al., 2008; Guenther et al., 2006). Most of these dependencies are included in MEGAN2.1 used here (Guenther et al., 2012) with exception of the emission response to the soil water status. The importance of this is reported by measurement studies, such as Pegoraro et al. (2004), which have shown a significant reduction of the emission flux under drought conditions. Thus, especially, in the light of global warming with more frequent droughts, the implementation of this dependence is considered as important here.

Also, the simulation of nitrous acid (HONO), which is the main precursor of OH in polluted regions and significantly affects $NO_x$ and ozone in the troposphere. However, current chemistry models underestimate the atmospheric fate of HONO (Gonçalves et al., 2012; Zhang et al., 2016). One potentially missing source are HONO soil emissions. Since measurements by Su et al. (e.g., 2011); Oswald et al. (e.g., 2013) have reported significant HONO emissions fluxes during the nitrification of soils comparable to soil nitrogen monoxide (NO) emissions, the model implementation of this source is deemed to reduce the

underestimation.

About 10-20 % of the global $NO_x$, a main $O_3$ precursor, is produced by lightning activity (Miyazaki et al., 2014). However, the comparison to measurements have indicated some model uncertainties (Miyazaki et al., 2014; Tost et al., 2007b). The here used parameterization by Grewe et al. (2001), in particular, assumes the same lightning activity over land and ocean. Tost et al. (2007b) report that this leads to an overestimated lightning activity over the ocean when comparing to observations.

Here, we present several sensitivity studies with the global atmospheric chemistry model ECHAM/MESSy (EMAC, described in Sec. 2) in order to assess the importance of the described processes. The model simulations are evaluated against ground-based ozone measurements (Sec. 3.1) and satellite retrievals of tropospheric $O_3$ (Sec. 3.2) and $NO_2$ (Sec. 3.3). In section 4-9, the sensitivity results are analysed assessing the global impact, importance and quality of each modification with respect to the observations. We discuss the remaining uncertainties of modelling $O_3$ and $NO_x$ in the troposphere (Sec. 10) and

conclude the study with an additional description of future developments.

## 2   Global modelling

The here applied Global Atmospheric Chemistry Model ECHAM/MESSy (EMAC) is a numerical climate modelling system which combines the Modular Earth Submodel System (MESSy, version 2.54.0; Jöckel et al., 2016) with the fifth generation European Centre Hamburg general circulation model (ECHAM5,version 5.3.02; Roeckner et al., 2003). MESSy contains

multiple submodels for among others physical and chemical processes and links multi-institutional computer codes to the basemodel. Thus, it provides a flexible infrastructure for coupling processes to build comprehensive Earth System Models (ESMs). In the present study we conducted simulations with EMAC at a resolution of T42L47MA ($2.8° \times 2.8°$, up to 0.01 hPa). The model dynamics have been weakly nudged towards by the assimilation of data from the European Centre for Medium-





range Weather Forecasting (ECMWF) (Jöckel et al., 2010) towards a realistic meteorology. The chemistry does not feed back

to the dynamics, resulting in the same meteorology for all simulations (QCTM mode; Deckert et al., 2011). The soil hydrology scheme (bucket model) of ECHAM5, which drives the surface fluxes and the related atmospheric variables, is initialised with an updated data set of field capacity from Hagemann and Stacke (2015). As vegetation information (Leaf Area Index: LAI) within the submodels SURFACE, DDEP, VERTEX, ONEMIS and MEGAN monthly data derived from the Moderate Resolution Imaging Spectroradiometer (MODIS) based on an algorithm by Knyazikhin et al. (1998) is used. This data set have

been aggregated by Klingmüller et al. (2017) based on a reprocessed version of MODIS LAI products by Yuan et al. (2011). The relevant atmospheric gas-phase chemistry in the troposphere and stratosphere is calculated within the submodel MECCA (Module Efficiently Calculating the Chemistry of the Atmosphere) using the Mainz Organic Mechanism (MOM; Sander et al., 2019), which contains an extensive oxidation scheme for isoprene (Taraborrelli et al., 2009, 2012; Nölscher et al., 2014), monoterpenes (Hens et al., 2014), and aromatics (Cabrera-Perez et al., 2016; Taraborrelli et al., 2021) excluding iodine and

mercury chemistry. The resulting system of ordinary differential equations (ODEs) represents 697 gaseous species and 2099 reactions. MECCA employs a sophisticated tagging system that allows for obtaining reaction rates from multiple reactions and combining them into a single tracer (Gromov et al., 2010). This system allows to obtain detailed tropospheric budgets of tracers. When considering the global $O_3$ budget, odd oxygen ($O_x$) is analysed to account for rapid cycling between species of the $O_x$ family. In the scope of this work, $O_x$ is defined as:

$$
\begin{aligned}
O_x \equiv\ & O + O_3 + NO_2 + 2 \times NO_3 + 3 \times N_2O_5 + HNO_3 \\
& + HNO_4 + ClO + HOCl + ClNO_2 + 2 \times ClNO_3 \\
& + BrO + HOBr + BrNO_2 + 2 \times BrNO_3 + PANs \\
& + PNs + ANs + NPs
\end{aligned}
\tag{1}
$$


where PANs are peroxyacyl nitrates, PNs are alkyl peroxynitrates, ANs are alkyl nitrates, and NPs are nitrophenols. For the troposheric $O_x$ budget presented in Table 3, the tropopause is defined in the extratropics using potential vorticity, whereas temperature lapse rates are used in the tropics (Jöckel et al., 2010).

The removal of trace gases and aerosol particles by clouds and precipitation is simulated by the SCAVenging submodel

(SCAV; Tost et al., 2006). SCAV calculates the transfer of species into and out of rain and cloud droplets using the Henry's law equilibrium, acid dissociation equilibria, oxidation–reduction reactions, heterogeneous reactions on droplet surfaces, and aqueous-phase photolysis reactions representing more than 150 reactions (Tost et al., 2007a).

The submodel DDEP provides the dry deposition of trace gases (Kerkweg et al., 2006) in which the dry deposition at the surface is determined based on the standard resistance-in-series model by Wesely (1989). Biogenic emissions of VOCs are

parameterised with an emission activity algorithm by the submodel MEGAN (Guenther et al., 2006) and soil emissions of NO are calculated according to an empirical model by Yienger (1995) in the submodel ONEMIS. Anthropogenic emissions are based on data from the RCP8.5 scenario performed for the fifth assessment report (AR5) of the Intergovernmental Panel on Climate Change (IPCC) (Lamarque et al., 2010), vertically distributed following Pozzer et al. (2009). For representing biomass burning emissions the Global Fire Assimilation System (GFAS) inventory is used which contains observed dry matter burn and





fire types (Kaiser et al., 2012). This integrates the emission factors and fire types for different trace gases by Andreae (2019) and Akagi et al. (2011). The production of $NO_x$ during lightning is parameterised with the scheme by Grewe et al. (2001) in combination with the Tiedtke-Nordeng convection scheme (Tost et al., 2007b). The scheme is based on the convective upward mass flux by Tiedtke (1989) and on the thickness of convective clouds. Vertically, the emission of $NO_x$ is distributed with the widely used C-shape profile following Pickering et al. (1998) (Grewe et al., 2001). The $NO_x$ production efficiency used in

the simulations is taken from Price et al. (1997). It assigns a production of 15.6 kg(N) per cloud-to-ground (CG) flashes and a factor of ten lower production for intra-cloud (IC) flashes and the flash frequency is scaled to 6.548 as in Jöckel et al. (2016). This yields an annual global production efficiency of 318 $mol(NO)\,flash^{-1}$ which agrees with the reported estimates of 250 – 400 $mol(NO)\,flash^{-1}$ of known studies (Gordillo-Vázquez et al., 2019; Miyazaki et al., 2014; Schumann and Huntrieser, 2007).

All in all, we performed one reference and five sensitivity simulations, covering the years 2008 and 2009 (2008 used for spin up). Finally, one simulation comprise the multiple model developments suggested in this study. An overview of the different simulations is given in Table 1. The detailed model development are motivated and explained in the following sections (4-9) each including an analysis of the global changes. The respective budgets of $O_x$ are given in Table 3.

## 3   Observational data

### 3.1   Station measurements from the Tropospheric Ozone Assessment report (TOAR)

For evaluating surface ozone, we use ground-based station measurements collected in the TOAR database (Schultz et al., 2017). The database comprises in-situ hourly data from almost 10,000 measurement sides around the globe which are available for the period 1970–2015. These have been collected from different ozone monitoring networks (e.g., Clean Air Status and Trends Network (CASTNET)) and data providers (e.g., Umweltbundesamt), harmonised and synthesised to enable an uniform

processing. The data were selected based on an extended quality control; e.g., sites where the measurement technique changed with time have been excluded. Data errors still remain but have been shown to have a minor impact (Schultz et al., 2017). The total uncertainty in modern ozone measurements is estimated with $< 2\,nmol\,mol^{-1}$ (Tarasick et al., 2019). The here used $O_3$ product is shown in Figure 1.

### 3.2   $O_3$ IASI satellite product

The satellite measurements of tropospheric ozone used here to evaluate the EMAC simulations are based on the observations by the Infrared Atmospheric Sounding Interferometer (IASI) instrument, a thermal infrared (TIR) Fourier transform spectrometer which measures the backwards radiation of the Earth's surface and of the lower atmosphere in a nadir-viewing geometry (Clerbaux et al., 2009). IASI operates on the Metop satellites surrounding the Earth on a polar, sun-synchronous orbit and crossing the Equator at around 9:30 a.m. and p.m. (local solar time). IASI provides $\sim 1.3$ million TIR spectra each day and

ensures a twice-daily near global coverage (Clerbaux et al., 2009). The vertical abundance and global distributions of ozone





are derived in near real time from the individual IASI spectra with the Fast Optimal Retrievals on Layers for IASI (FORLI) algorithm version 20151001 (Hurtmans et al., 2012). The quality of the IASI profile product is ensured by the application of a series of specific quality flags rejecting poor spectral fits, data with poor vertical sensitivity and cloud contaminated IASI scenes (Wespes et al., 2018). The IASI tropospheric column of ozone is defined as ranging between the ground and 300 hPa. This

range also allows limiting the influence of stratospheric $O_3$ on the retrieved tropospheric column, while still including the layers of maximum sensitivity of IASI in the troposphere (Wespes et al., 2017). The total statistical error from the retrievals for the tropopsheric column is estimated to to about 5-20 % (Hurtmans et al., 2012; Wespes et al., 2016; Boynard et al., 2018). For the here used partial column up to 300 hPa, Boynard et al. (2018) have reported a negative bias by $O_3$-FORLI in the mid-latitudes and in the tropics (11-13 % and 16-19 %, respectively) compared to ozonesonde data. In order to account for the vertical

sensitivity of IASI to $O_3$ measurements, the averaging kernels associated with each retrieved $O_3$ profiles were considered for the model-to-satellite comparisons. In this context, we used the MESSy submodel SORBIT (Jöckel et al., 2010) to sample the complete EMAC vertical profiles (in volume mixing ratio; VMR) at the time and location of the IASI measurements. The co-located EMAC profile was first interpolated to the FORLI-O3 pressure grids and then converted into column profile. Then, the altitude-dependent sensitivity of IASI has been taken into account by applying the FORLI-O3 averaging kernels on

the co-located EMAC profile following the formalism of Rodgers (2000). The tropospheric $O_3$ columns were then calculated from the convoluted profiles between the ground and 300 hPa. Since several IASI observations usually fall into a given model $2.8° \times 2.8°$ grid box for a given day, the interpolation/convolution of the corresponding EMAC profile was repeated separately for each IASI measurements falling in that model grid box on that day (see, e.g., Schultz et al., 2018; Rosanka et al., 2021b). A daily mean value of $O_3$ tropospheric column was eventually calculated for that grid box by averaging all the columns obtained

from the repeated smoothing of the EMAC profile. The application of the IASI averaging kernels to the EMAC ozone profiles, returns simulated tropopsheric O3 as seen by IASI. The resulting product is shown in Figure 2.

### 3.3 NO₂ OMI satellite product

Due to the potential impact of the model developments on the tropospheric level of $NO_2$ which is an important precursor of tropospheric ozone, we use the $NO_2$ tropospheric columns measured by the Ozone Monitoring Instrument (OMI). OMI

has been launched on board NASA's Earth Observing System Aura satellite in July 2004. The nadir-viewing near-UV-visible (400-470 nm) spectrometer measures the Earth's backscattered radiance and solar radiance in a Sun-synchronous polar orbit, crossing the equator at 13:40 local solar time (Zara et al., 2018). From this, the total slant density column (SCD) of a trace gas (concentration integrated along the light path between the earth surface and the satellite) is calculated using the differential optical absorption spectroscopy (DOAS) method based on the Beer-Lambert law. The stratospheric part of the SCD is estimated

by the Data assimilation transport model TM5 (http://www.qa4ecv.eu/ecv/no2/main/strato) which is then subtracted from the total SCD to retrieve the tropospheric SCD. Dividing by the tropospheric air mass factors (AMFs) yields the tropospheric vertical column density (VCD). For a harmonised comparison to account for OMI sensitivity, the averaging kernel provided by the QA4ECV product has been applied whereas the original (TM5) a priori profiles from OMI QA4ECV product have been replaced with that from the EMAC simulation. The $NO_2$ VCD product used in this study has been developed by the Quality





Assurance for Essential Climate Variables (QA4ECV) project consortium with an improved spectral fitting algorithm which yields the smallest uncertainties ($0.8 \times 10^{15}$ molecules cm$^{-2}$) comparing the previously existing NO$_2$ products. In general, surface and atmospheric properties (e.g. cloudiness, solar zenith angle, snow cover, large aerosol close to the surface) largely affect the vertical sensitivity of the instrument. Therefore, the data used here was filtered with discarding those corresponding to processing error flag, cloud radiance fraction > 0.5, solar zenith angle > 80 °, snow covered land and ocean (> 10%

snow cover) and the ratio of tropospheric airmass factor and geometric airmass factors < 0.2 (description of the flags in the documentation). By that, however, cloud-free conditions are preferred for tropospheric VCDs and negative biases occur, especially over polluted regions. General uncertainties of the tropospheric NO$_2$ retrievals can be related to the separation of tropospheric from stratospheric SCD (Zara et al., 2018). Beirle et al. (2016) have shown that the separation of the stratospheric and tropospheric SCD in the calculation of the NO$_2$ satellite retrievals leads to a general uncertainty between 0.1 and $0.2 \times 10^{15}$

molecules cm$^{-2}$ (5-10 %) which is negligible in high polluted areas but plays a role in moderately polluted areas. In comparison with ground-based measurements acquired by MAX-DOAS the OMI QA4ECV products show a negative bias of 1-4 $\times 10^{15}$ molecules cm$^{-2}$ (Compernolle et al., 2020; Kumar et al., 2020).

In order to ensure the representativity during comparison with the simulations, data has been sampled along the satellite track by the MESSy submodel SORBIT (Jöckel et al., 2010). Then, this has been integrated over the location dependent tropopause

level given in the QA4ECV product. The data is only analysed if at least 10 values exist. The resulting product is shown in Figure 3.

## 4 The role of water vapour in radical reactions

Water vapour has a high abundance in the troposhere, in particular in the lowest layers where increasing levels are predicted during global warming. Besides its role as the most important greenhouse gas warming the Earth's atmosphere, water vapor is

known to form complexes with radicals (Buszek et al., 2011). These complexes modify the reaction kinetics and often act as catalyst enhancing the fate and the speed of the reactions (e.g., Zhang et al., 2019; Kumbhani et al., 2015; Buszek et al., 2011). Since hydroxy and peroxy radicals are involved in the formation of photochemical species such as O$_3$, the water-complexes influence the budgets of these trace gases as suggested by, e.g., Butkovskaya et al. (2009).

### 4.1 Modified kinetics of NO$_x$

In MOM, the reaction rate constant of HO$_2$ + NO forming NO$_2$ and OH is represented originally according to Burkholder et al. (2015) as:

$$k_{HO_2+NO} = 3.3 \times 10^{-12} \times \exp\left(\frac{270}{T}\right). \tag{2}$$

In order to allow the formation of water complexes, simulated in EMACh2o, this is replaced by $k_{1w}$, which is defined as:

$$k_{1w} = k_0 - k_{2w} \tag{3}$$





where $k_0$ is the recommended value by Sander et al. (2003). Additionally to the $NO_2+OH$ channel, another $HNO_3$ formation channel, represented with the reaction rate constant $k_{2w}$, is considered explicitly for the first time in this study and added to EMACh2o. The importance of this formation channel has been clearly indicated by e.g., Righi et al. (2015).

$$k_{2w} = \frac{(\beta \cdot k_0 \cdot (1 + 42 \cdot \alpha)}{(1 + \alpha) \cdot (1 + \beta)} \cdot \frac{1}{2}$$

where $\beta$, also used in Cariolle et al. (2008), is based on Butkovskaya et al. (2009) and is recommended by JPL (J. B. Burkholder and Wine, 2020). The detailed calculation of $\alpha$ is given in Butkovskaya et al. (2009); Gottschaldt et al. (2013). Following the results of Duncianu et al. (2020), the 2-fold lower reaction rate $k_{2w}$ is used here. The potential role of water complexes in the kinetics of organic peroxy radicals has been indicated in a few studies (Kumbhani et al., 2015; Clark et al., 2008, 2010). Recently, Xing et al. (2018) proposed that substituted $RO_2-$water complexes during the oxidation of methacrolein might have a unity alkyl nitrate yield when reacting with NO and estimated a lower limit for the equilibrium constant for the complexation. Although independent supporting evidence for the enhancement of the alkyl nitrate yield under humid conditions is warranted, its effect for all substituted $RO_2$ is implemented here based on the data derived for MACRO2 by Xing et al. (2018). Recently, Amedro et al. (2020) presented experimental evidence that water molecules are six times more efficient than $N_2$ in quenching the $HO-NO_2$ association complex in the $OH + NO_2$ reaction. The effect of water vapour can enhance the reaction rate by up to 15 %. Combined with experimental data valid for dry conditions by Amedro et al. (2019), a new parameterization for atmospheric modelling was implemented in the EMAC model and is also used here.

## 4.2 Impact on the tropospheric ozone chemistry

Allowing the formation of water complexes in radical reactions have significant impacts on the chemistry of $O_x$ as seen in Table 3. The chemical production and loss terms of $O_x$ in the troposphere are decreased by about 20 % compared to the reference simulation (EMACref), which brings the model in better agreement with the recent multi-model estimates of 4500-5200 $Tg\,a^{-1}$ chemical production and 4000-4800 $Tg\,a^{-1}$ loss (Young et al., 2018). As indicated by Butkovskaya et al. (2009), who assumes that the $HNO_3$ channel at 50 % humidity is as important as the $HNO_3$ production from $OH + NO_2$, the enhanced formation of $HNO_3$ from the reaction of $HO_2$ with NO including water complexes is dominant in many regions (Fig. 4a). This reduces the $NO_2$ yield from the $HO_2 + NO$ reaction as well as OH, and thus the oxidation capacity of the atmosphere (Gottschaldt et al., 2013). The additional produced $HNO_3$, however, is deposited faster than converted into $NO_2$ and transported to the Southern oceans (decrease of $NO_2$) (Fig. 4b). Consequently, less $O_3$ is formed by $NO_2$ photolysis which is strongest on the southern hemisphere continents. In higher polluted regions like North America, Europe and East Asia, the modified $HO_2+NO$ reaction affect significantly the $HO_2$ yield (Fig. 4c), also reported by Butkovskaya et al. (2009). Due to the counter-balance with a slight increase of $NO_2$ the impact on $O_3$ is smaller than on the SH. The modified $RO_2+NO$ kinetics play a minor role as it can be estimated from the changed dry deposition flux of the produced alkyl nitrates ($\sum AN$; see Equation A1). Enabling the formation of water complexes increases the $\sum AN$ dry deposition significantly from 0.35 to 0.96 $Tg(N)\,a^{-1}$ which brings the EMAC model in better agreement with the estimate by Archer-Nicholls et al. (2021, their Table 8). The global burden of the two ANs, $i-C_3H_7NO_3$ and $n-C_3H_7NO_3$ increases from 15.3 to 65.7 and 10.1 to 43.8



Gg, respectively, which is much higher than the estimates by Khan et al. (2015, Tab. 5). The higher burden in EMAC can be attributed to the rapid hydrolysis of isoprene nitrates (Vasquez et al., 2020) not being represented in the here used model

simulation, which is known to reduce the burden and dry deposition of nitrates. From the dry deposition estimate a conclusion for $O_3$ can be drawn since the $RO_2$/NO reaction produce either one alkyl nitrate or two ozone molecules (major branch) (Day et al., 2003). In fact, the change of the $\sum AN$ dry deposition flux corresponds to 3.3 $Tg\,a^{-1}$ less $O_3$ removal, which is minor relative to the absolute $O_3$ deposition (Tab. 3). The contribution of the $RO_2$/NO reaction to the change of the $O_x$ production is with 7 % much smaller than the 70 % contribution by the $HO_2$/NO reaction. Overall, the inclusion of water complexes

decreases the mismatch between modelled and measured surface ozone by 10-15 % towards a small remaining bias of $\pm$ 4 $nmol\,mol^{-1}$ (Fig. 11b,f). The highest mismatch occur in sub-urban coastal grid boxes where an accurate representation of chemical concentrations is notoriously challenging since the steep gradients of mixing ratios from land to ocean (stronger during daytime) are not resolved in models (Fiore et al., 2002). Also, the annual mean tropospheric column ozone is decreased by 10-20 % due to the modified kinetics (Fig. 13b). Thus, the bias of EMAC relative to IASI is significantly reduced (between

40°S and 40°N), as shown in Figure 12a and Figure 12b. The overall tropospheric $O_3$ burden, simulated by EMACh2o, of 347 Tg (-12 %, Tab. 3) agrees much better with the recent multi-model mean estimate of 340 $Tg\,a^{-1}$ of $O_3$ burden and the observed values, e.g. global tropospheric burden of 324 Tg derived from the IASI-FORLI observations (Gaudel et al., 2018), where the used tropopause definition, is however different than the one used here. Also, Righi et al. (2015) have reported a significant reduction of the ozone bias in the troposphere due to the inclusion of the $HNO_3$ production from $HO_2+NO$.

Contrarily, the model underestimation of tropospheric $NO_2$ in comparison to OMI is strengthened by EMACh2o (Fig. 16b).

## 5   The role of weather for dry deposition

Dry deposition of trace gases to vegetation depends on several meteorological variables which often determines the variability of ozone levels (Wong et al., 2019; Val Martin et al., 2014; Kavassalis and Murphy, 2017). In fact, the relation to air temperature is well known and has been reported by many studies (e.g., Jarvis, 1976; Hogg et al., 2007). The importance of linking ozone

uptake with the atmospheric water demand has been shown by e.g. Fares et al. (2012); Hogg et al. (2007) to be essential to capture the daily depression in the afternoon. In the light of global warming with higher temperatures and more frequent droughts, these links become increasingly important and the integration in models is desirable (Andersson and Engardt, 2010; Wong et al., 2019).

### 5.1   The advanced dry deposition scheme

The uptake of trace gases in EMAC is based on the 'big-leaf approach' and calculated by the means of multiple resistors which represent the different spheres of the soil-vegetation-atmosphere system (e.g., Wesely and Hicks, 2000; Wesely, 1989). Both methods are commonly used in Earth system models due to its simplicity as reported in the review by Clifton et al. (2020) or as applied by e.g. Val Martin et al. (2014); Zhang et al. (2003). In this study, an extension of the general framework described by Kerkweg et al. (2006) is used in the EMACddep simulation. The extended scheme includes additional dependencies of the





stomatal uptake on temperature and atmospheric water demand as multiplicative stress factors. Also, an explicit calculation of

the uptake at the plants' cuticle (wax covering of the leaves) has been incorporated as well as the consistent usage of the Leaf

Area Index (LAI), which is the basic vegetation information of the scheme, as described by Emmerichs et al. (2021).

### 5.2  Impact on the $O_x$ dry deposition and chemistry

The inclusion of the revised dry deposition scheme by Emmerichs et al. (2021) enables a proper cuticular uptake leading to a

higher deposition over vegetated areas, which is amplified by the stomatal response in moderate climate (optimal temperature:

25°C, humid air). Balancing effects occur in dry and hot climate as well as in the Tropics (highest LAI) due to the oversim-

plification of dry deposition in the reference scheme with a LAI of 1 (Emmerichs et al., 2021). Hence, the impact shows a

spatial and temporal variation where $O_3$ dry deposition increases with the growing of vegetation on the northern hemispheric

continents up to 20 %. In this area, surface ozone mixing ratios decrease by 5 to 10 %. If day- and nighttime values are studied

separately the change by EMACddep plays relatively a more important role during nighttime over most vegetated areas as

shown in Figure 5 for boreal summer. Since the newly implemented cuticular uptake does not rely on sunlight, contrasting to

the stomatal pathway. Therefore, also the balancing effects described above does not occur during nighttime. The revised dry

deposition decreases the overestimation of EMAC with respect to the TOAR observation data during day and night by 5-10 %,

where during the day the bias is generally lower with a remaining discrepancy of only up to $\pm$ 6 $\mathrm{nmol\,mol^{-1}}$ (Figs.  11c, g).

The remaining mismatch in Europe and USA during daytime could be explained by the here under-represented NO emissions

from soils, most important in boreal summer and autumn. Since the reaction with NO have been reported as relevant chemistry

sink for $O_3$ during the day (e.g., Fares et al., 2012).

Globally, the dry deposition flux of $O_x$ and $O_3$ is estimated in the reference simulation (EMACref) with 825.2 and 780.2

$\mathrm{Tg\,a^{-1}}$, respectively (Tab. 3). This is in agreement with the multi-model range of 700-1470 $\mathrm{Tg\,a^{-1}}$ reported by Young et al.

(2018); Wild (2007); Hu et al. (2017). The inclusion of the revised dry deposition scheme increases the global annual dry

deposition of $O_3$ and $O_x$ slightly by 28 and 26 $\mathrm{Tg\,a^{-1}}$, respectively (Tab. 3). Hence, EMAC's prediction of $O_3$ dry deposition

gets closer to the multi-model mean of 1000 $\mathrm{Tg\,a^{-1}}$ reported by (Young et al., 2018). The representation of $O_3$ dry deposition

could be further improved by explicitly representing $O_3$ deposition to soil, which several measurement studies report to be an

important pathway (e.g. Fares et al., 2012; Stella et al., 2011). Moreover, we know that due to a dry bias (too dry soil and too

high temperature) in the Amazon basin the dry deposition (Hagemann and Stacke, 2015) is highly underestimated in this area,

very important for dry deposition (Emmerichs et al., 2021).

### 6  Impact of the isoprene emission modelling

Isoprene is the most important biogenic emitted VOC accounting for 50-70 % of the total emissions (Guenther et al., 2006;

Sindelarova et al., 2014). Common studies use the MEGAN model (Guenther et al., 2012) which estimates the emission fluxes

with multiple activity factors accounting for different environmental responses. The here used MEGAN2.1 covers several of

these responses but lacks the representation of drought stress, since in conditions of less precipitation and dry soil (i.e. drought)





the emissions of isoprene are affected where the response depends on the duration of the drought (Ferracci et al., 2020). Overall, a significant decline of isoprene emissions have been observed due to droughts (Pegoraro et al., 2004; Ferracci et al., 2020). The representation of this response is expected to become more important in the light of global warming with an increasing

number of droughts (Samaniego et al., 2018).

### 6.1 The drought stress activity factor

Accounting for the drought/soil moisture response in models yields a reduction of up to 70 % for the isoprene emissions depending on the choice of the wilting point (e.g. Huang et al., 2015; Guenther et al., 2012; Sindelarova et al., 2014). To overcome this uncertainty the sensitivity simulation EMACisop applies a multiplicative factor in place of the commonly used

soil moisture stress factor based on the soil wetness which was calculated for every month of the time period 2005-2014 (Jiang, 2018) by Jiang et al. (2018). Since all VOC emissions in MEGAN are based on the same approach the stress factor is applied to all species.

### 6.2 The global impact

Isoprene is mainly emitted in the tropical rain forests of Brazil, Central Africa and Indonesia (e.g. Sindelarova et al., 2014;

Guenther et al., 2006; Müller et al., 2008; Henrot et al., 2017), such as simulated by EMAC (Fig. A3). The usage of the drought factor (seasonal means in Fig. A2), which have been shown to improve the accuracy of the model in capturing drought and non-drought periods (Jiang et al., 2018), leads to a decrease of the emission flux in the emission regions of about 20 $\mathrm{mg\,m^{-2}\,s^{-1}}$ in boreal summer (Fig. 6a). The major reaction partner OH, thus shows an increase which cause a reduction of $O_3$ at the surface (Fig. 6b). These results are similar to those presented by Jiang et al. (2018) applying the drought activity factor

in CAMS-Chem. For the comparison to the observed tropospheric $NO_2$ column this effect is only important in relative terms. Hence, also the difference of the EMAC and the IASI tropospheric ozone column is not influenced significantly. Globally, the annual isoprene emission flux is reduced from 508.2 to 347.8 $\mathrm{Tg(C)\,a^{-1}}$ (-32 %) due to the usage of the drought stress activity factor. These annual emission values are generally in agreement with the estimated range of 260 to 566 $\mathrm{Tg(C)\,a^{-1}}$ for global isoprene emissions which, however depends on the period, the model resolution and the input data (Arneth et al., 2011;

Weng et al., 2020). For 2008, Young et al. (2009) estimated 401 $\mathrm{Tg(C)\,a^{-1}}$ ($2.5° \times 3.75°$) whereas Henrot et al. (2017) gave a flux estimate of 417 $\mathrm{Tg(C)\,a^{-1}}$ for 2000-2012 ($1.8° \times 1.8°$). Both, however, do not account for the response of isoprene emissions on the soil moisture state. Including the soil moisture stress factor for isoprene, Weng et al. (2020) gave a global annual isoprene emission of 333 $\mathrm{Tg(C)\,a^{-1}}$ for 2014-2017 with a similar model resolution to the one used here. Taken this as the most comparable estimate (Jiang et al. (2018) give no global estimate), we conclude overall that EMACisop represents

reasonably the global isoprene emissions while it overcomes the major uncertainty related to the parameterization of soil moisture stress for isoprene emissions (Jiang et al., 2018). However, we note that the range of model estimates for the global isoprene emission is large due to the existing uncertainty in modelling (Arneth et al., 2011).





## 7   Soil emissions of HONO as a missing global source

Nitrous acid (HONO) is an important OH source, especially in the morning, and thus it impacts significantly the $HO_x$ and $O_3$
budgets (Monks, 2005; Sörgel et al., 2011). However, current models tend to underestimate HONO levels, which is attributed
to an incomplete knowledge of HONO sources (e.g. Li et al., 2010). Recently, soil emissions have been shown to close this
gap (Su et al., 2011; Yang et al., 2020) with important implications for $HO_x$ and $O_3$. Emissions of HONO from biological
soil crusts have been estimated to be $0.69\,\mathrm{Tg(N)\,a^{-1}}$ (Porada et al., 2019). However, a global estimate of HONO from soil
bacteria is not available yet.

### 7.1   The flux parameterization

The latest released version of EMAC does not represent HONO emissions from soil. In order to address its importance on
tropospheric $O_x$, this source is included in the EMAChono simulation. Since the soil emissions of HONO and NO are mech-
anistically similar (Oswald et al., 2013; Su et al., 2011), the emission flux of HONO is calculated by mapping the measured
HONO/NO ratio from different boreal field samples by Oswald et al. (2013) to the four main land cover types (crops, forest,
grassland, savanna) (Tab. A1).

### 7.2   The global impact

The simulated soil emission fluxes of HONO range up to $10\,\mathrm{ng(N)\,m^{-2}\,s^{-1}}$ with the highest emissions at crops (90 %)
and woody savanna (71 %) (Fig. A1). This matches the lower end of the fluxes reported by Su et al. (2011) and Oswald
et al. (2013), and thus accounts only partly for the missing measured fluxes of about $1\text{-}1000\,\mathrm{ng(N)\,m^{-2}\,s^{-1}}$ (for mixed-layer
heights of 100 to 1000 m) reported by Ren et al. (2010) and Li et al. (2010). The reason for this low fluxes is likely the
general underestimation of the NO soil emission by the Yienger&Levy scheme (by 24 %) (Steinkamp et al., 2009). Globally,
the model yields an annual HONO emission flux of $7\,\mathrm{Tg(N)\,a^{-1}}$ from soil. The increase of surface HONO, highest in Central
Africa and South and East Asia, leads to enhanced mixing ratios of $NO_2$ over these areas. This brings the HONO/$NO_x$ ratio
(Fig. 7) closer to the measured values of 0.13-0.53 (Chai et al., 2019) whereas in Australia and Central Africa the simulation
overestimates the reported ratio. Globally, EMAChono simulates an annual HONO/$NO_x$ ratio of 0.06 at the surface which is
a significant enhancement compared to the estimate of EMACref (ratio: 0.007) but overestimates the calculated global ratio of
0.02 by Elshorbany et al. (2012). Regarding the tropospheric chemistry, OH (as a photolysis product of HONO) is increased
significantly over the southern continents (40 %) to a slight extent over Europe, Asia, USA (10-20 %) which is shown in Figure
8a. The OH increase at the surface on the southern hemisphere comply with the reported negative bias of models (Naik et al.,
2013). The inter-hemispheric disparity of OH (Lelieveld et al., 2016; Patra et al., 2014), however, is not influenced significantly
by EMAChono. The OH increase reduces the isoprene level at the surface over most regions. Since HONO also represents a
source of $NO_x$ its increase enhances the annual mean tropospheric $NO_2$ column by up to $0.5 \times 10^{15}\,\mathrm{molecules\,cm^{-2}}$ (60 %)
which is the highest over Australia and Central Africa. But this has a minor impact on the EMAC/OMI bias. The enhanced
$NO_2$ and HONO levels add up to a small relative increase of tropospheric ozone (up to 6 %) in the Tropics. All these changes





influence the global chemical production and loss of $O_x$. In the planetary boundary layer (PBL), the $O_x$ chemical production and loss is increased by 12 % and 6 %, respectively, due to the modification which yields a net increase of $O_x$ by chemistry (details in Tab. 3. This increase is also shown in Figure 8 for ground-level ozone over the SH continents (no observations available). This might improve the known underestimation of ground-level ozone in the Southern Hemisphere (Young et al., 2013). Regarding the whole troposphere, also a net increase is seen, but this change is minor ( 4.5 %).

Overall, this development represents one of the first implementation of an additional HONO source, the emission from soil in a global model and shows realistic values. However, the sparsity of measurements used for mapping the observed ratio to a global distribution introduces some uncertainties. The measured emission fluxes have an error of $\sim$ 1 % for grassland/savanna and 4 % for forests/crops. Also, the simplicity of the scheme, which does not account for the reported dependency of soil HONO fluxes on soil water content (Oswald et al., 2013) due to the structure of the existing NO soil parameterization, intro-
duces uncertainties.

## 8 NOₓ emission by lightning

The production of $NO_x$ by lightning activity accounts for 10-20 % of the global $NO_x$ budget (Miyazaki et al., 2014). Kang et al. (2020) even report a 30 % contribution to surface ozone, comparable to the anthropogenic $NO_x$ source, in the Eastern US summer. But the comparisons to measurements point to the relevant modelling uncertainties of 1.4 $\mathrm{Tg(N)\,a^{-1}}$ (Miyazaki
et al., 2014; Tost et al., 2007b). Namely, the schemes diverge on the choice of IC /CG $LNO_x$ ratio and thus on the vertical distribution of $LNO_x$ emission (Gordillo-Vázquez et al., 2019), which causes the wide range of estimated global flash rate and emission values.

### 8.1 The lightning NOₓ parameterization

The $LNO_x$ scheme by Grewe et al. (2001), used in EMACref as the current standard in EMAC, overpredicts lightning over the
ocean since the flash activity is treated equally over land and oceans. Furthermore, the data assimilation of multiple satellite data by Miyazaki et al. (2014) suggests that the C-shape based parameterizations, such as the Grewe scheme, overestimate the peak source height by up to 1 km over land and the tropical western Pacific. In contrast, the scheme by Price and Rind (1992) (P&R), which is commonly used e.g. for the CMIP6 simulations (Griffiths et al., 2021), stands out by representing lightning over land and ocean with distinct flash frequencies. Also, it is described as robust in space and time. Hence, the P&R scheme
is applied here in the sensitivity simulation EMAClnox.

### 8.2 The global impact on tropospheric NO₂ and O₃

Figure 9 shows the global annual lightning frequency estimated by the simulations EMACref and EMAClnox. The comparison with the observed global annual flash rates of the Optical Transient Detector (OTD)/Lightning Imaging Sensor (LIS) (Gordillo-Vázquez et al., 2019, Fig. 1a) and (Miyazaki et al., 2014, Fig. 2a) shows a general agreement of the global distribution.
Over oceans the lightning activity is generally weak, but it still represents an important source of $NO_x$ in the absence of





anthropogenic sources with a flash rate of up to 3 flashes $km^{-2} a^{-1}$ (Fig. 9b). The Grewe parameterization used in EMACref calculates too high flashes over the free ocean due to the applied vertical C-shape profile as it has been indicated by Tost et al. (2007b). This is most pronounced in the western Pacific. Applying the common P&R parameterization in EMAClnox decreases the $NO_x$ lightning emissions over the Indian and the tropical Pacific oceans towards a better agreement with the

observed values. Miyazaki et al. (2014) claimed that the dependency in the P&R scheme is too weak over the ocean. The flash rates over the regions, such as Indonesia and South America however, are higher than the observations. The overestimation over South America has been also reported by Miyazaki et al. (2014). In literature the P&R scheme is criticised for linking the cloud top height and the electrification only indirectly (Tost et al., 2007b). The changes of the lightning activity correspond to a 20-30 % increase of the tropospheric $NO_2$ column over land and a 40-60 % decrease over oceans. This improves the bias

between EMAC and OMI data over the tropical oceans and some continental emission regions like North Africa and Central U.S (Fig. 16c). The changes of $NO_x$ impact the tropospheric ozone column in the Tropics leading to a subsequent 5-10 % decrease of the annual mean $O_3$ over the tropical Pacific (towards a lower bias) and a small increase over the tropical continents (Figs. 12c, 13c). Globally, the P&R scheme yields 423 $mol(NO) flash^{-1}$ (+32 %) whereas the common estimates range from 200 to 600 $mol(NO) flash^{-1}$ (Miyazaki et al., 2014; Gordillo-Vázquez et al., 2019; Nault et al., 2017; Marais et al., 2018).

This corresponds to a global annual $LNO_x$ emission of 5.7 $Tg(N) a^{-1}$ (as in EMACref) which agrees well with the reported range of estimates (Tab. 4).

As demonstrated here, the usage of the commonly applied P&R scheme improves significantly the $NO_2$ bias between EMAC and OMI as well as the tropospheric $O_3$ discrepancy to IASI over the tropical Pacific Oceans.

## 9 Advanced representation of tropospheric ozone

The previous sections demonstrate the importance of the different model developments applied in this study, which is summarised in Table 2. We combine the implementation of these developments in the simulation EMACmulti. An exception is the inclusion of the $H_2O$ complexes, for which the effect of water in the $RO_2/NO$ reaction is neglected in EMACmulti due to the sparse evidence of kinetic data. Also, the global impact of this modification on the $O_x$ chemistry is minor (Sec. 4).

### 9.1 Global impact on tropospheric ozone and its precursors

Among the different sensitivity simulations EMACh2o has the largest impact on ozone at the surface and in the troposphere (up to 300 hPa). Although the revised dry deposition scheme influences surface ozone significantly in some regions (Sec. 5), its overall impact in EMACmulti is minor (Fig. 10). The flux is indeed mainly driven by the decreased background concentration which arise from the modified kinetics in EMACh2o. Thus, the discrepancy between EMACmulti and TOAR surface ozone in Europe and the United States as shown in Figure 11d and Figure 11h corresponds to the comparison with EMACh2o. The

remaining EMAC/TOAR bias of $\pm 4$ $nmol mol^{-1}$ lays for most regions within the measurement uncertainty.

Considering the troposphere below 300 hPa, two other effects contribute additionally to the overall change. On the one hand, the ozone increase caused by the inclusion of HONO soil emissions (Sec. 7) counterbalances the impact of the water inclusion




in the Tropics and extra-Tropics. Over the tropical oceans, the tropospheric $O_3$ column is significantly reduced by the usage of the P&R lightning $NO_x$ scheme (Sec. 8). This leads to a significant reduction of tropospheric ozone by about 15 % (Fig. 13d).
The resulting bias of up to 6 DU represents an improvement against the former reported EMAC bias towards satellite retrievals 15 DU reported by Jöckel et al. (2016). The remaining mismatch exceeds the uncertainties on the IASI retrieved tropospheric $O_3$ column of 5-20 %. The remaining $O_3$ bias over the oceans is probably associated with an underestimated ozone loss by halogens (Sherwen et al., 2016) and $HO_x$-chemistry in clouds (Rosanka et al., 2021b).

Considering the total tropospheric $O_3$ column, the changes by EMACmulti (Fig. 15b) are even stronger (down to -30 %).
Here a large reduction is seen over the southern polar regions, in contrast to the comparison of the smoothed model data (Fig. 13). This is due to the lower sensitivity of IASI over the polar regions (low brightness temperature) characterised by the applied averaging kernel (see Sec. 3.2). Figure 14 show that the induced changes reach up the tropopause at all latitudes, the largest in the extratropical southern hemisphere and in the tropical upper troposphere. Besides the important impact on tropospheric chemistry, which can be fully discussed here, the $O_3$ decrease modify the radiative forcing of tropospheric ozone.
In fact, Stevenson et al. (2013, and references therein) have estimated a change of $\sim$40 mW m$^{-2}$ total radiative forcing per one DU. The radiative effect of pertubations per unit mass of ozone is maximum in the tropical upper troposhere where our revised model predicts changes of -10 % or lower (Riese et al., 2012).

Tropospheric $NO_2$ overall is decreased by EMACmulti (Fig. 16d). The decreased $NO_2$ levels from EMACh2o are balanced and partly compensated by the increase of tropospheric $NO_2$ due to EMAChono, the strongest in Central Africa and Australia
(Sec. 7). Globally most important are the reduced $NO_2$ levels over the oceans which lower the model overestimation in the tropical Pacific and Indian Ocean but cause a negative bias in the Northern Atlantic Ocean. An underestimation over South Africa and East Asia exceeding the OMI bias ($1 \times 10^{15}$ molecules cm$^{-2}$) remains. Since South Africa is a region with high biomass burning emissions the bias which is highest in boreal summer and autumn (biomass burning seasons) might be due to an under-predicted fire emission factor in this area. In contrast, the underestimation over East Asia can be attributed to
anthropogenic emissions which is the highest $NO_x$ source in this region. Overall, however, the noted discrepancies are within the relative uncertainties of $NO_x$ emissions (Sec. 10).

Considering the calculated $O_x$ budgets, the chemical production and loss estimates by EMACmulti match better the most recent multi-model estimates of 4500-5200 Tg a$^{-1}$ chemical production and 4000-4800 Tg a$^{-1}$ loss where EMACref gives to high estimates. In contrast, the global dry deposition estimate by EMACmulti is much lower than the recent multi-model
mean estimate of 1000 Tg a$^{-1}$ (Young et al., 2018). This might point to an under-represented dry deposition (e.g. at soil) in the model (Sect. 5). But it has to be noted that the calculated global dry deposition flux depends on the background concentration of ozone and therefore the decreased $O_x$ dry deposition is a consequence of the reduced net production (Tab. 3).

## 10   $NO_x$ emissions and the uncertainty

$NO_x$ is the major precursor of tropospheric ozone driving the chemical production and loss (Monks, 2005). In low and moderate
polluted areas like South America and Africa, the dominant sources are soil emissions and biomass burning from savanna fires





(Velders et al., 2001). The comparison of the estimated soil NO emissions by EMAC to literature values at a similar resolution and time period (Tab. 4) indicates a general underestimation of soil NO (25 % of global $NO_x$) in EMAC by the Yienger&Levy scheme, as also reported by Steinkamp et al. (2009). For all landcover types except in the tundra and rainforest, the emission are under-represented due to the pulsing parameterization. Biomass burning emissions calculations are quite uncertain (Andreae, 2019; Andreae and Merlet, 2001) as seen from the listed literature values in Table 4. The general uncertainty of the GFAS inventory is around 30 %, but increases for species like $NO_x$ in the important fire regions Siberia, Central Africa and Indonesia where insufficient ground measurements lead to a lack of spatial and variable information (Andela et al., 2013). A further high uncertainty is likely due to undetected small fires (>100 hectare) whose fraction Ramo et al. (2021) estimated to 80-100 % over Africa. The MODIS instruments on the Aqua and Terra satellites, which are among others the origin of the GFAS inventory, lack the observations of small fires. Especially, towards the equator where the overpasses of the polar orbiting satellites reduce to two days. Moreover, the fire radiative power is underestimated by GFAS in the Tropics since the average of daily observation does not account for the high oscillation during day in these areas (Andela et al., 2013). This likely reasons the absolute difference between the annual mean tropospheric $NO_2$ column of EMAC and OMI over southern Africa, as it have been also reported by Andela et al. (2013). In this region, the most widerspread fires occur during boreal summer and autumn (see NASA Earth Observatory, last access: 8 June 2021) where the reported EMAC/OMI $NO_2$ bias is highest. A further contributing reason for the $NO_2$ underestimation could be that peat fires which also occur in this region and emit low amounts of $NO_x$ and high amounts of VOCs are hardly estimated in bottom-up approaches, such as GFAS (Krol et al., 2013).

It has been shown that anthropogenic $NO_x$ emissions have an uncertainties of at least 20 % (Solazzo et al., 2021; Ding et al., 2017). As these emissions account for roughly 65 % of the total $NO_x$ emissions (Pozzer et al., 2012), anthropogenic sources can have a large contribution on the overall uncertainties, especially over the North Hemisphere where most of these emissions are located.

Lightning activity is the major source of $NO_x$ over remote oceans accounting for 10-30 % of the global emissions (Miyazaki et al., 2014; Kang et al., 2020). The regional distribution of lightning is highly uncertain, as it has been shown by Tost et al. (2007b) applying different convection and lightning parameterizations in the EMAC model. Especially, in the tropics the representation of cumulus convection is challenging (Miyazaki et al., 2014). Further uncertainty is attributed to the non-linear $NO_x$ photo-chemistry which can only be captured in high-resolution models. Namely, $NO_2$ is converted rapidly to $NO_y$ and $NO_z$ (e.g. $HNO_3$) in remnant $NO_x$ plumes after thunderstorms (Nault et al., 2017). Observational constraints take this non-linear $NO_x$ photo-chemistry (plume chemistry) into account and are used to adjust the NO production per flash and thus tune emissions parameterizations. However, this plume chemistry is a sub-grid scale process for most global atmospheric models and is commonly not parameterized. Representation of the plume chemistry for $NO_x$ by lighting has been shown to yield significant changes in the $O_3$ levels predicted by a global model (Gressent et al., 2016). In addition, the recent study by Brune et al. (2021) has reported relevant $HO_x$ production by lightning activity which also would explain a high $NO_x$-to $NO_z$ conversion.



## 11 Conclusion and outlook

Many processes determining tropospheric ozone are affected by meteorology (Kavassalis and Murphy, 2017; Porter and Heald, 2019). Here, we have implemented or applied important features in EMAC influencing the relationship between ozone and weather. Namely, the formation of water-complexes in reactions of hydroxyl and hydroperoxy radicals with nitrogen oxides is enabled affecting the tropospheric ozone chemistry. For dry deposition at vegetation we apply a parameterization extended with additional meteorological stress factors for the stomatal uptake and a weather-dependent explicit formulation of cutic-

ular uptake. In addition, soil emissions of nitrous acid, the major precursor of the OH radical, are represented for the first time in EMAC. Also, important for tropospheric ozone, gaining more relevance in the light of global warming, is the here considered drought-dependence of biogenic volatile organic compound emissions. Finally, we investigate the production by lightning activity, a relevant source of nitrogen oxides, using a parameterization with a better distinction of land and ocean. A detailed analysis of the separate impacts by means of the tropospheric odd oxygen budget and the subsequent comparison

with ozone measurements at the surface (TOAR) and for tropospheric columns (IASI) have been conducted. In addition, the impacts on nitrogen dioxide have also been investigated with respect to the OMI QA4ECV product. The inclusion of the model developments overall reduces the bias between simulated and measured ozone abundances. In fact, the comparison with TOAR data at ground level results in a remaining mismatch of $\pm 4 \, \mathrm{nmol \, mol^{-1}}$ (during day). This mainly arise from the inclusion of $HO_2$-water complex in the $HO_2 + NO$ reaction and the enhanced cuticular deposition. This represents an improvement against

the reported ACCMIP multi-model ensemble bias to TOAR data (Young et al., 2013). The model overestimation of ozone in the troposphere (up to 300 hPa) is decreased by about 75 % (averaged over the year), most important over East Asia and the tropical Pacific ocean, towards a remaining positive bias of 2-6 DU. The changes arise from the inclusion of $HO_2$-water complex reducing nitrogen oxides as well as lower lightning $NO_x$ emissions over the oceans. Globally, these effects are partially counteracted by the inclusion of HONO emissions from soil. The remaining model bias for tropospheric ozone relative to IASI

observations can be further reduced by representing a more comprehensive $HO_x$-chemistry in clouds (by 1-2 DU; Rosanka et al., 2021b) and iodine chemistry (Sherwen et al., 2016). These expected reductions, however, will be partially compensated by $NO_x$-recycling resulting from heterogeneous chemistry leading to $ClNO_2$ (Riedel et al., 2014) and HONO (Benedict et al., 2017), which are not included in the EMAC model yet. The overall reduction of tropospheric $NO_2$ mainly leads to solving the model overestimation over the tropical oceans. Overall, the bias is within the uncertainties of the $NO_x$ emissions.

The NO soil emission representation in EMAC, which do not present a contiguous dependence on soil moisture and temperature yet, will be updated with the parameterization of Hudman et al. (2012). Such parameterization also represents pulsing of the emissions following dry spells and N-inputs from chemical fertilizer and atmospheric N-deposition, yielding 34 % more annual global soil emissions of nitrogen oxide with a higher contribution from fertilisation. For emissions of biogenic volatile organic compounds the implementation of MEGAN3 (Jiang et al., 2018) in EMAC will lead to some improvements, by allow-

ing at least a consistent online calculation of the drought emission activity factor, which we showed to have a non-negligible impact on total ozone column in the model. Finally the use of a more up-to-date anthropogenic emissions database for the studied year would improve the results against the observational dataset, especially over polluted regions. For example, the





EDGAR (Emissions Database for Global Atmospheric Research, version 4.3.2) do have 8 % larger global anthropogenic $NO_x$ emissions for the year 2010 than the one used here, implying a much larger difference at regional level. The here investigated weather-dependent processes such as dry deposition will be considered in simulations at much higher spatial resolution for studying air pollution during extreme heat and drought events, which are known to increase in frequency during global warming.

*Code availability.* The Modular Earth Submodel System (MESSy) is continuously further developed and applied by a consortium of institutions. The usage of MESSy and access to the source code is licenced to all affiliates of institutions which are members of the MESSy Consortium. Institutions can become a member of the MESSy Consortium by signing the MESSy Memorandum of Understanding. More information can be found on the MESSy Consortium Website http://www.messy-interface.org. The code used in this study is included in the current devel branch of the MESSy repository

*Data availability.* The simulation results are archived at the Jülich Supercomputing Centre (JSC) and are available on request. The TOAR $O_3$ data is available upon request. The IASI $O_3$ data processed with FORLI–$O_3$ v20151001 can be downloaded from the Aeris portal at http://iasi.aeris-data.fr/O3/ (last access: 25 February 2021). The OMI QA4ECV data is freely avaialble for download at http://www.qa4ecv.eu/ecv/no2-pre/data.





## Appendix A: Alkyl nitrates

In this study, the sum of all alkyl nitrates is defined following the MOM definition (following Sander et al., 2019) as:

$$\sum AN = BZBIPERNO3, PROPOLNO3, LBUT1ENNO3,$$
$$BZEMUCNO3, IBUTOLBNO3, ISOPBNO3, ISOPDNO3,$$
$$LAPINABNO3, BPINANO_3, IC3H7NO3, NC3H7NO3,$$
$$LMBOABNO3, C514NO3,$$
$$C614NO3, BZEMUCNO_3, C6H5CH2NO3,$$
$$TLBIPERNO3, TLEMUCNO3 \tag{A1}$$

*Author contributions.*  T.E. and D.T. designed the study and discussed the model developments which were implemented by T.E.. S.R. wrote the script for calculating the budget of odd oxygen and the related technical description. V.K. contributed with the comparison of EMAC $NO_2$ data with OMI QA4ECV data product. B.F. and C.W. processed the IASI data, applied the averaging kernels and contributed to the analysis of the EMAC/IASI comparisons. A.P. contributed to the discussion of the emissions. T.E. performed the EMAC simulations, the data analysis, prepared the figures and wrote the manuscript with the help of all co-authors.

*Competing interests.*  The authors have no competing interests

*Acknowledgements.*  The work described in this paper has received funding from the Initiative and Networking Fund of the Helmholtz Association through the project "Advanced Earth System Modelling Capacity (ESM)". The content of this paper is the sole responsibility of the author(s) and it does not represent the opinion of the Helmholtz Association, and the Helmholtz Association is not responsible for any use that might be made of the information contained. The authors gratefully acknowledge the usage of computing time on the
JURECA supercomputer at the Jülich Supercomputing Centre (JSC). . We further thank Niklas Selke for preparing the TOAR dataset and Martin Schultz for helpful discussions on the use of the TOAR data. IASI is a joint mission of Eumetsat and the Centre National d'Etudes Spatiales (CNES, France). The authors acknowledge the Aeris data infrastructure for providing access to the IASI data and ULB–LATMOS, in particular Daniel Hurtmans, for the development of the retrieval algorithms. The research at ULB is funded by the Belgian Federal Science Policy Office (BELSPO) and the European Space Agency (ESA) under the ESA–BELSPO Prodex arrangement (IASI.FLOW) and
the Satellite Application Facility on Atmospheric Composition Monitoring (AC–SAF).





**Table 1.** EMAC simulations

| Simulation | development |
| --- | --- |
| EMACref | Settings as described in Section 2 |
| EMACddep | Revised dry deposition scheme according to Emmerichs et al. (2021) |
| EMACisop | Drought emission activity factor for isoprene (in MEGAN) by Jiang et al. (2018) . |
| EMAChono | Implementation of HONO soil emissions (in ONEMIS) |
| EMACh2o | Modified kinetics of $HO_2+NO$, $NO+RO_2$ and $OH+NO_2$ reactions taking into account water complexes |
| EMAClnox | Lightning $NO_x$ scheme by Price et al. (1997): Scaling factor for flash frequency $= 9.29$, annual global lighting $NO_x$ emission $= 5.7 \ Tg(N) \, a^{-1}$ (as in REF). |
| EMACmulti | EMACddep, EMAChono, EMACisop, EMAClnox, modified $HO_2+NO$ and $NO_2$ OH reaction |



**Table 2.** Summary of model developments and advances

| Simulation | Significance of the implementation | Level of implementation and uncertainty | Improvement regarding the observation data (TOAR[a]/IASI/OMI)[b] |
|---|---|---|---|
| EMACh2o | inclusion of important $H_2O$ complexes & $HNO_3$ formation from the $HO_2/NO$ reaction | modified kinetics based on literature, implemented $RO_2/NO$ kinetics are speculative | decrease of $\Delta O_3$(EMAC-TOAR) by up to 10 $nmol\,mol^{-1}$ decrease of $\Delta O_3$(EMAC-IASI) by up to 5 DU $\pm 0.5, \times 10^{15}$ molecules $cm^{-2} \Delta NO_2$(EMAC-OMI) |
| EMAClnox | distinction of lightning $NO_x$ production over land and ocean | scaled to lower production efficiency, uncertainty of P&R reported in Tost et al. (2007b) | no impact on $\Delta O_3$(EMAC-TOAR) $\pm$ 3 DU $\Delta O_3$(EMAC-IASI) $\pm\ 0.5, \times 10^{15}$ molecules $cm^{-2}$ $\Delta NO_2$(EMAC-OMI) |
| EMACddep | add important deposition pathway and sensitivity to meteorology | empirical model, parameters aren't space-dependent | decrease of $\Delta O_3$(EMAC-TOAR) by 5 $nmol\,mol^{-1}$ $\pm$ 2 DU of $\Delta O_3$(EMAC-IASI) $\pm 0.1\ , \times 10^{15}$ molecules $cm^{-2}$ $\Delta NO_2$(EMAC-OMI) |
| EMAChono | missing additional HONO source, first global implementation | limited by one observation data set and four landcover types | up to + 7 $nmol\,mol^{-1}$ $\Delta O_3$(EMAC-TOAR) no improvement for $\Delta O_3$(EMAC-IASI) up to +0.2, $\times 10^{15}$ molecules $cm^{-2} \Delta NO_2$(EMAC-OMI) |
| EMACisop | First inclusion of important drought dependence for isoprene in EMAC | (offline) input data set | no impact on $\Delta O_3$(EMAC-TOAR) decrease of $\Delta O_3$(EMAC-IASI) by up to 2 DU no impact on $\Delta NO_2$(EMAC-OMI) |

[a]covers the United States and Europe
[b]Range of annual mean change





**Table 3.** $O_x$ budget main terms for the reference simulation (EMACref) in the troposphere and absolute changes induced by the sensitivity simulations. This includes chemical production (69 %: $HO_2 + NO$, 19 %: $CH_3O_2 + NO$, 12 %: $RO_2 + NO$), chemical loss (45 %: $O^{1D}) + H_2O$, 32 %: $HO_2 + O_3$, 13 %: $O_3 + OH$, dry deposition (95 %: $O_3$, 4 %: $HNO_3$), Scavenging (44 %: $HNO_3$, 43 %: $O_3$) and Stratosphere-Troposphere exchange (STE)

| | EMACref | ΔEMACh2o | ΔEMACddep | ΔEMACisop | ΔEMAChono | ΔEMAClnox | ΔEMACmulti |
|---|---|---|---|---|---|---|---|
| **Sources** [$Tg\,a^{-1}$] | | | | | | | |
| Chemical production[a] | 5550.6 | -1099.5 | -8.3 | -259.0 | +282.4 | -22.8 | -1092.0 |
| STE | 508.9 | +94.1 | +2.4 | +15.6 | -16.5 | -4.3 | +91.1 |
| **Sinks** [$Tg\,a^{-1}$] | | | | | | | |
| Chemical loss[b] | 4990.8 | -872.3 | -31.8 | -219.7 | +203.6 | -19.5 | -938.9 |
| Dry deposition | 825.2 | -116.8 | +28.4 | -19.3 | +44.9 | -21. | -56.2 |
| Scavenging | 243.5 | -16.7 | -2.6 | -5.1 | +17.3 | -0.5 | -5.1 |
| Net $O_x$ | 0 | +3.0 | -0.3 | -28.6 | +1.1 | -3.8 | -0.8 |
| $O_3$ **burden** [Tg] | 394.0 | -47.2 | -2.4 | -10.6 | +8.6 | -5.3 | -57.1 |
| $O_3$ **lifetime** [days] | 23.7 | +1.3 | -0.1 | +0.3 | -0.5 | -0.2 | +0.6 |

[a]Fraction of the chemical production in the PBL: 28.2-33.1 %
[b]Fraction of the chemical loss in the PBL: 24.9-25.8 %

**Table 4.** Global annual $NO_x$ and HONO emissions [$Tg\,a^{-1}$] by single studies and the third IPCC assessment report (TAR; Ehhalt et al., 2001)

| type/source | anthropogenic[a] $NO_x$ | lightning $NO_x$ | biomass burning $NO_x$ | soil $NO_x$ | soil HONO | biomass burning HONO |
|---|---|---|---|---|---|---|
| EMAC | 41.7 | 5.7 | 3.7 | 6.4 | 7.0 | 0.5 |
| single studies | 36.5-42.2[b] | 4.2-7.6[c] | 6.4-8.0[d] | 6.2-9.6[e] | | 1.0[f] |
| TAR | 33.7 | 5.0 | 5.6 | 7.1 | | |

[a]aircraft emissions included
[b]42.2 (Feng et al. (2020): 0.5°x0.5°, 2015); 40.0 (Cubasch and Winther (2013): 2009); 36.5 (McDuffie et al. (2020): 0.5°x0.5°, 2009)
[c]6.3 Miyazaki et al. (2014): 2.8°x2.8°, 2007); 4.5 (Jöckel et al. (2016): 2.8°x2.8°, 2009), 5.9 ± 1.7 (Marais et al. (2018): 2°x2.5°, 2006)
[d]6.4 (Feng et al. (2020): 0.25°x0.25°, 2015); 8.0 Andreae (2019), based on estimated emission factors and biomass burning emissions)
[e]7.6 (Miyazaki et al. (2017): 2.5°x2.5°, 2005-2014); 6.2, 9.6 (Hudman et al. (2012): 2°x2.5°, 2006); 9.0 (Stavrakou et al. (2013): 2°x2.5°, 2007)
[f]only estimate by Theys et al. (2020)





**Table A1.** Mapping scheme from categories in Oswald et al. (2013) to The International Geosphere Biosphere Programme (IGBP) land cover scheme (Belward, 1992)

| No. | IGBP category | soil samples |
| --- | --- | --- |
| 1 | forests | S1 -S4 |
| 2 | crops | S10-15, S17 |
| 3 | woody savanna | S6/7 |
| 4 | grassland | S5, S8/9 |





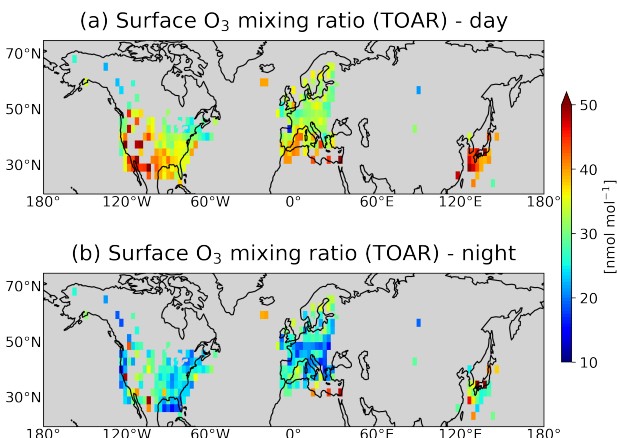

**Figure 1.** Annual mean surface $O_3$ mixing ratio during day (a) and night (b) by TOAR for the year 2009.

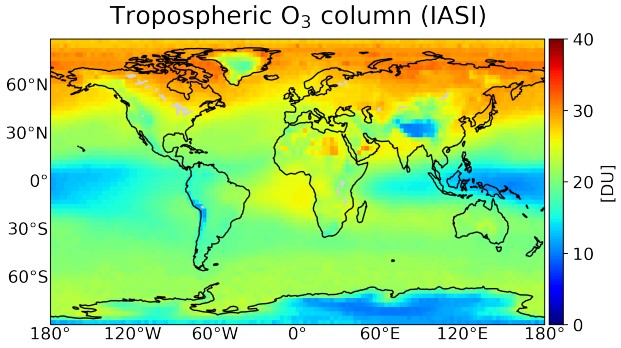

**Figure 2.** Annual mean tropospheric $O_3$ column by IASI for the year 2009.

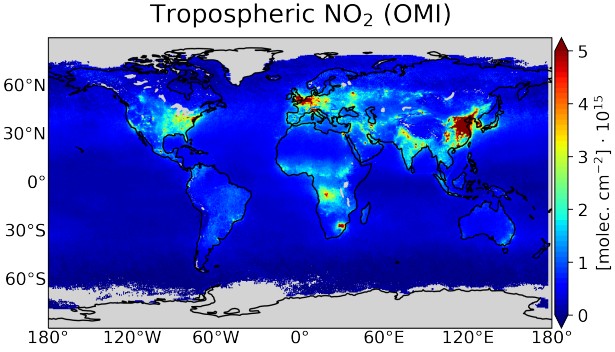

**Figure 3.** Annual mean tropospheric $NO_2$ by OMI for the year 2009.





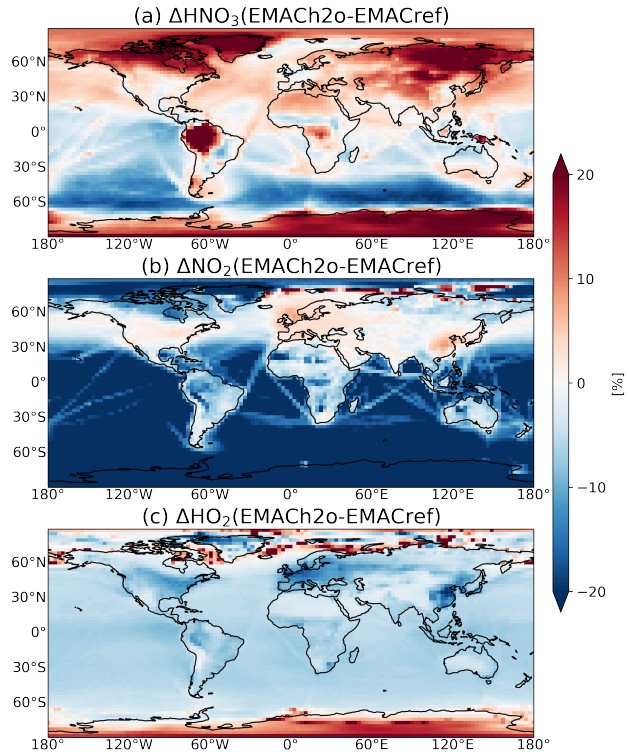

**Figure 4.** Relative difference of the annual mean mixing ratio of surface $HNO_3$ (a) and $NO_2$ (b) comparing EMACref and EMACh2o in 2009.

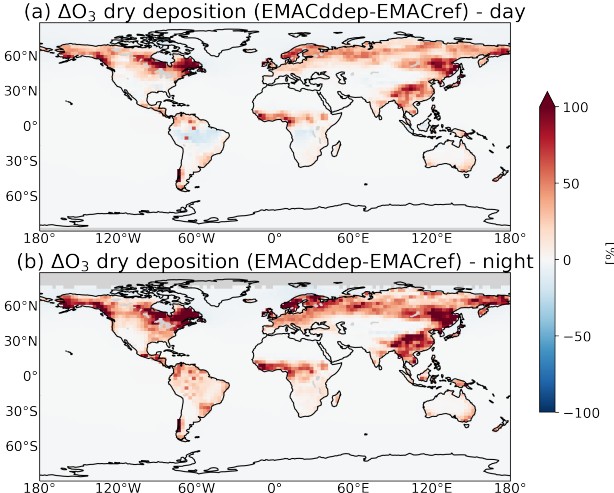

**Figure 5.** Relative difference (EMACddep-EMACref) of the 2009 (boreal) summer mean ozone dry deposition during day (a) and during night (b).



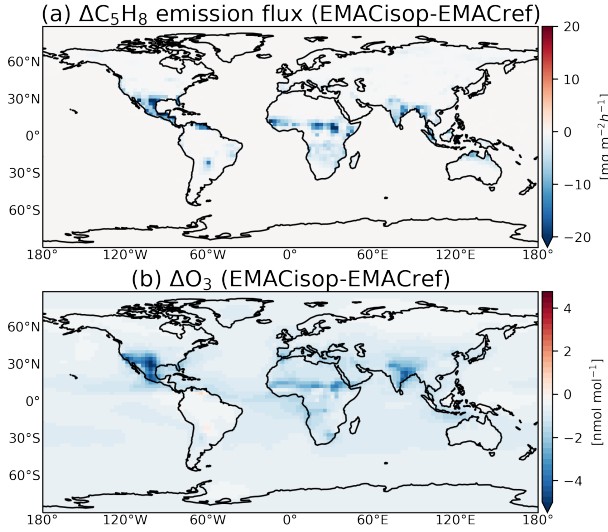

**Figure 6.** Absolute difference (EMACisop-EMACref) of the 2009 (boreal) summer mean isoprene emission flux (a) and surface ozone mixing ratio.

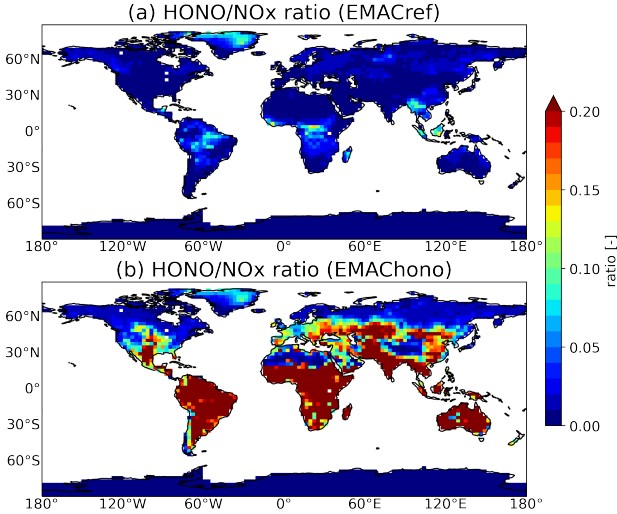

**Figure 7.** Ratio of the annual mean HONO to $NO_x$ mixing ratio predicted by EMACref (a) and EMAChono (b) at the surface in 2009.





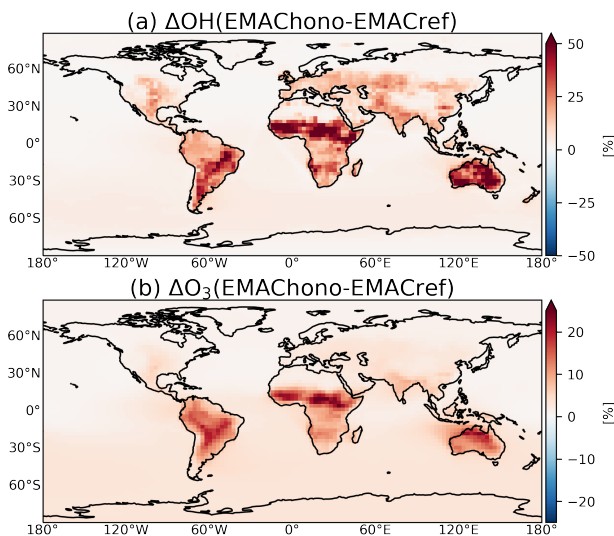

**Figure 8.** Relative change (EMAChono-EMACref) annual mean of OH mixing ratio near the surface (a) and $O_3$ mixing ratio (b) in 2009.

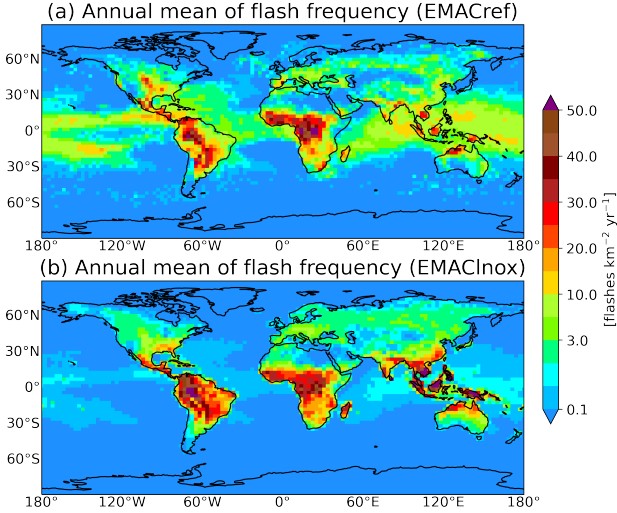

**Figure 9.** Annual mean flash frequency predicted by EMACref (a) and EMAClnox (b) in 2009.





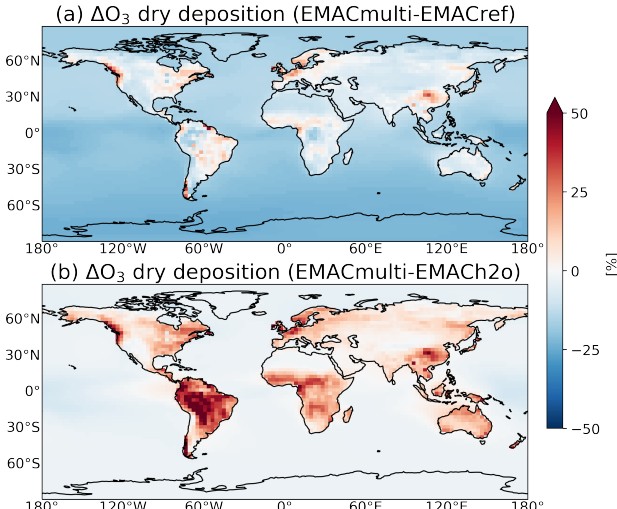

**Figure 10.** Annual mean relative difference of the $O_3$ dry deposition flux between EMACref and EMACmulti (a) and between EMACh2o and EMACmulti (b) in 2009.

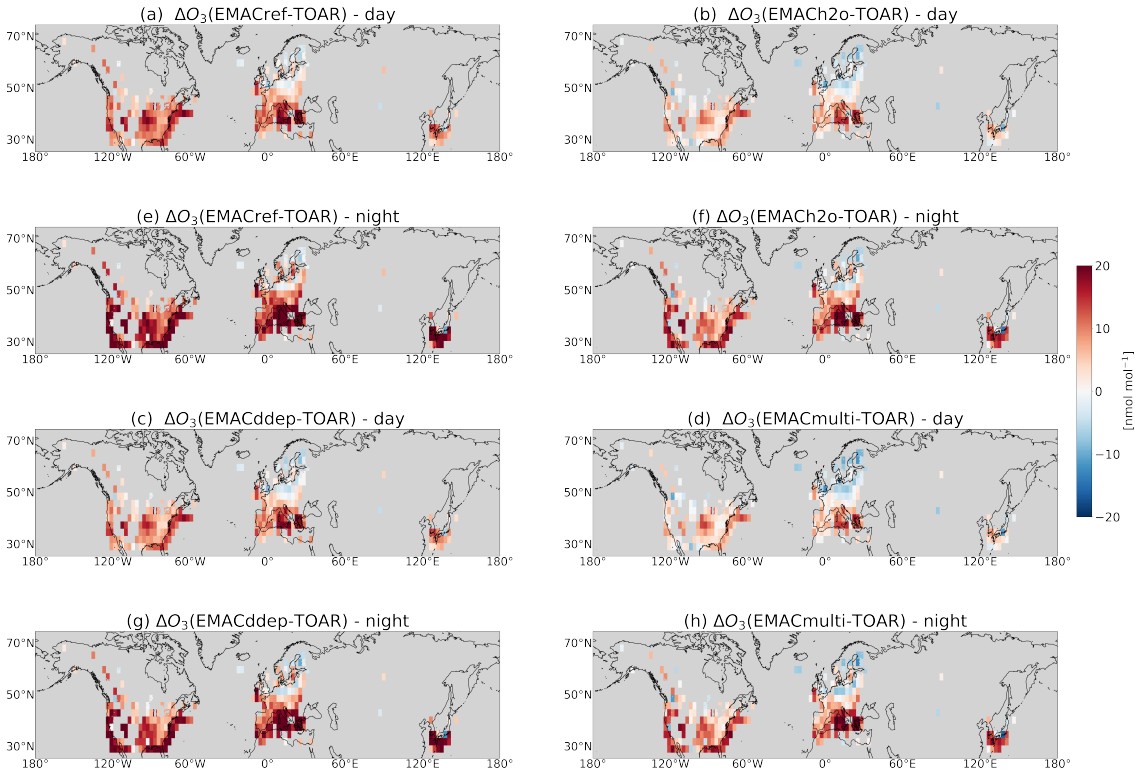

**Figure 11.** Annual mean absolute bias of the surface $O_3$ mixing ratio between EMACref and TOAR (a, e) and the absolute difference of EMACh2o (b, f), EMACddep (c, g) and EMACmulti (d, h) towards EMACref during day and during night in 2009, respectively.





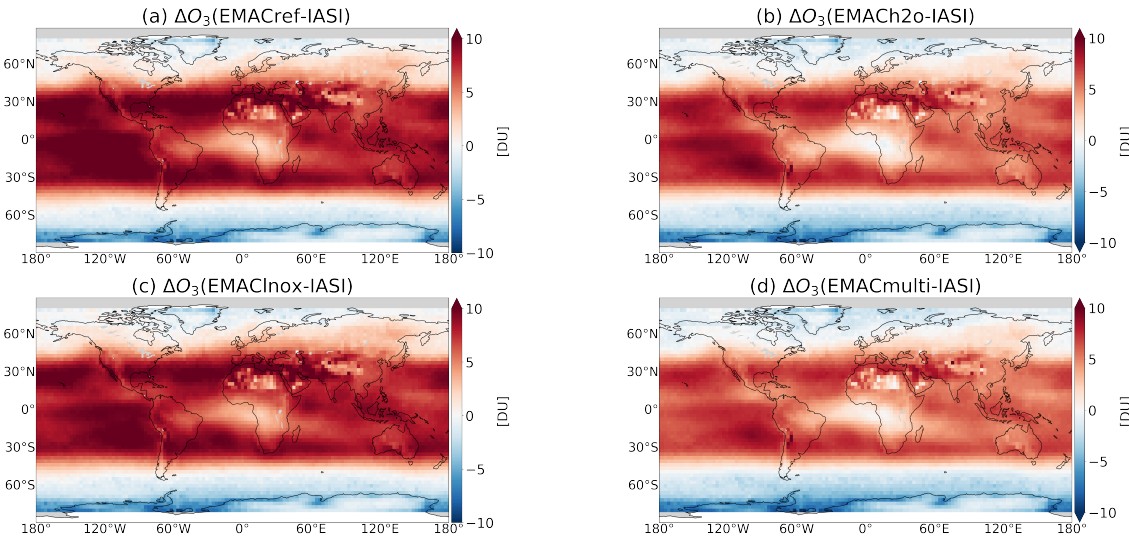

**Figure 12.** Annual mean bias of tropospheric $O_3$ (up to 300 hPa) of EMACref (a), EMACh2o (b), EMAClnox (c) and EMACmulti (d) towards IASI in 2009. The data have bben smoothed by applying the entirety of the IASI averaging kernel

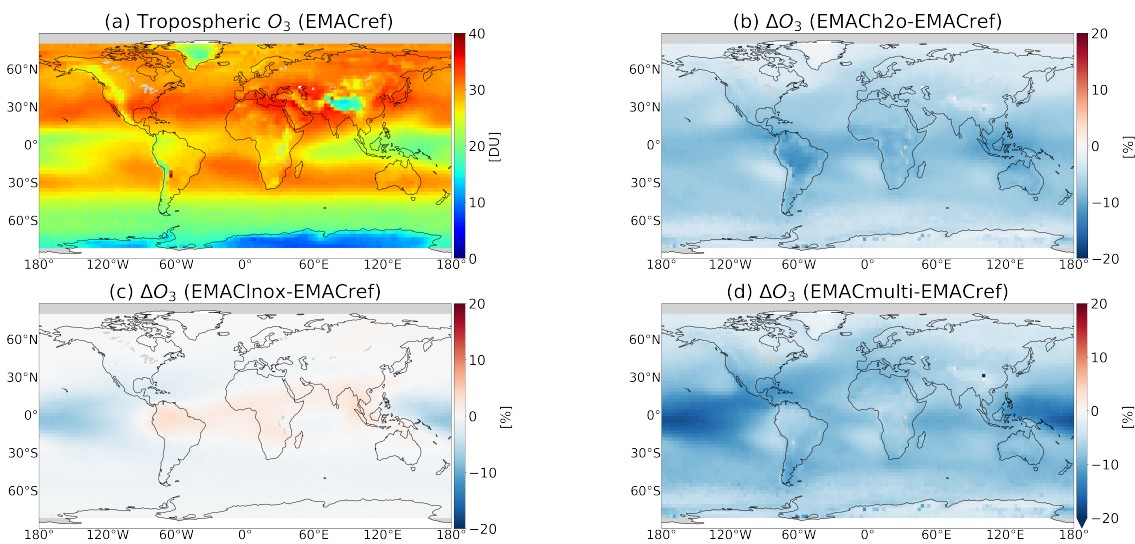

**Figure 13.** Annual mean tropospheric $O_3$ (up to 300 hPa) of EMACref (a) and the relative changes of EMACh2o (b), EMAClnox (c) and EMACmulti (d) towards EMACref in 2009, respectively. The data have been smoothed by applying the entirety of the IASI averaging kernel.





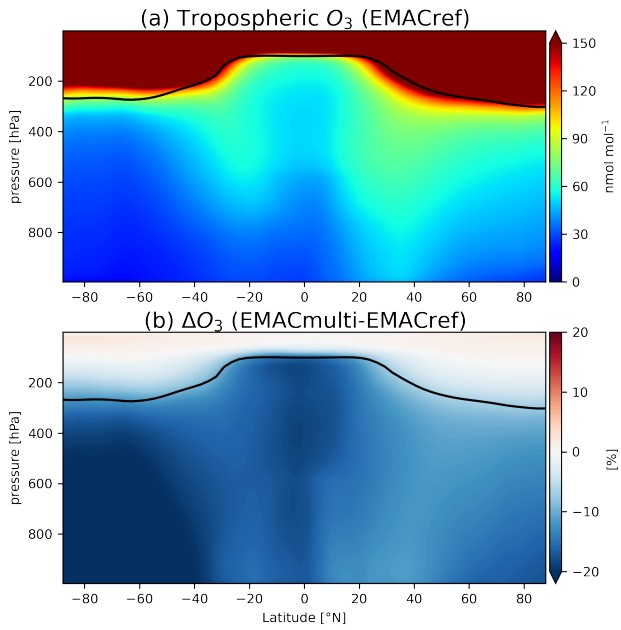

**Figure 14.** Mean zonal O₃ mixing ratios of EMACref (a) and the relative changes of EMACmulti towards EMACref in 2009.

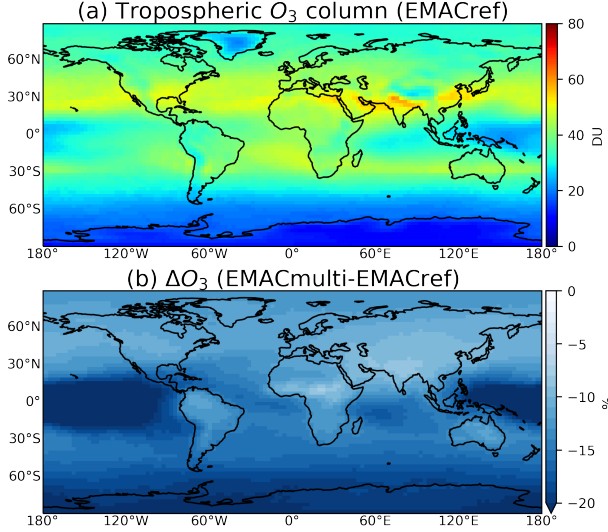

**Figure 15.** Annual mean tropospheric O₃ column of EMACref (a) and the relative changes of EMACmulti towards EMACref in 2009.





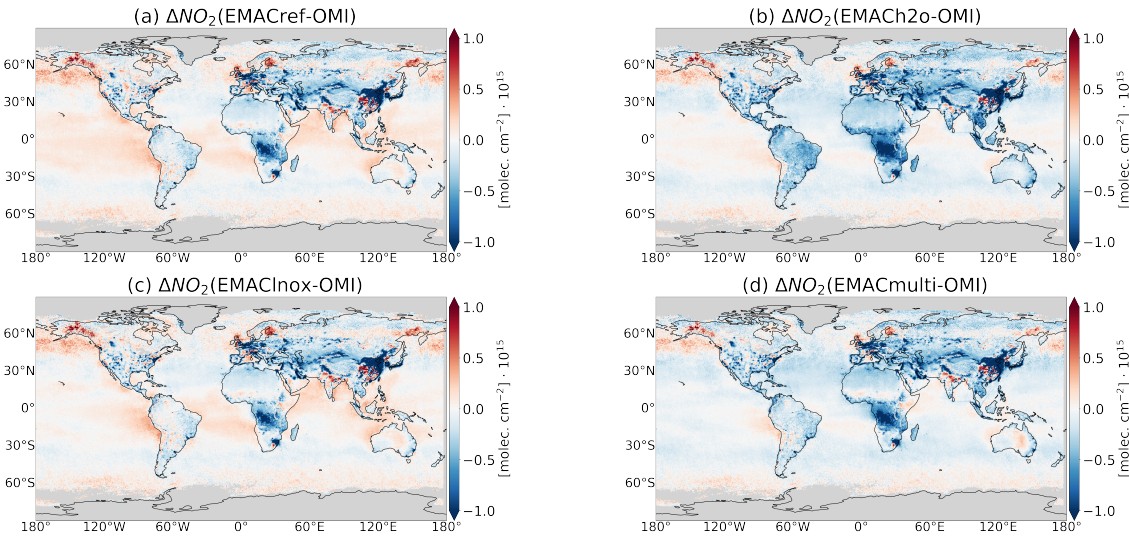

**Figure 16.** Annual mean bias of tropospheric $NO_2$ of EMACref (a), EMACh2o (b), EMAClnox (c) and EMACmulti (d) towards OMI in 2009. The data have been smoothed by applying the QA4ECV averaging kernel (up to the tropopause)

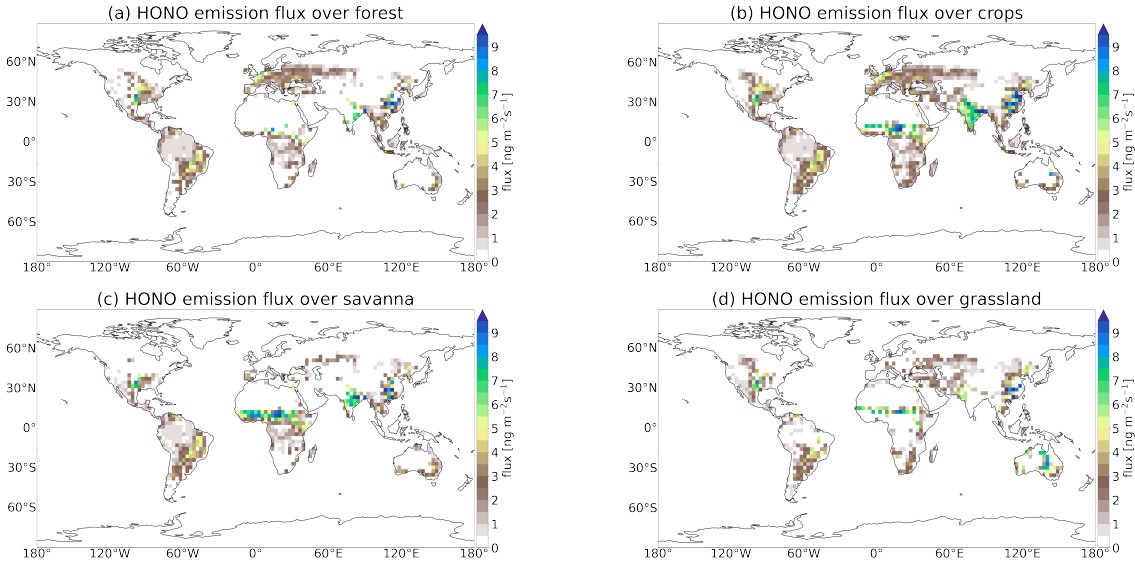

**Figure A1.** Annual mean (2009) of HONO emission fluxes at forest (a), crops (b), woody savanna (c) and grassland (d)





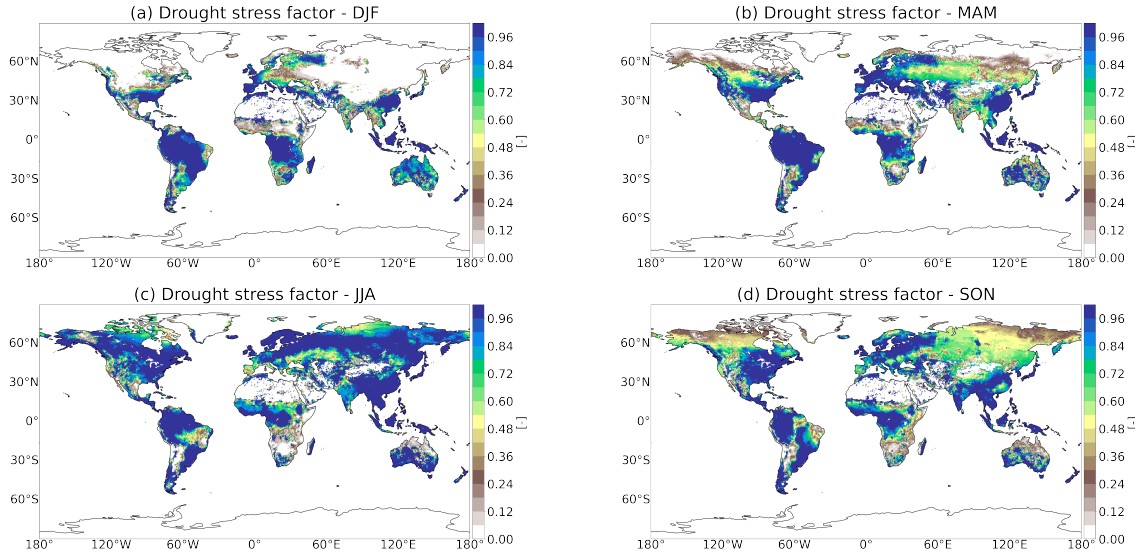

**Figure A2.** Seasonal means of isoprene drought activity factor in boreal winter (DJF) (a), spring (MAM) (b), summer (JJA) (c) and autumn (SON) (d) 2009

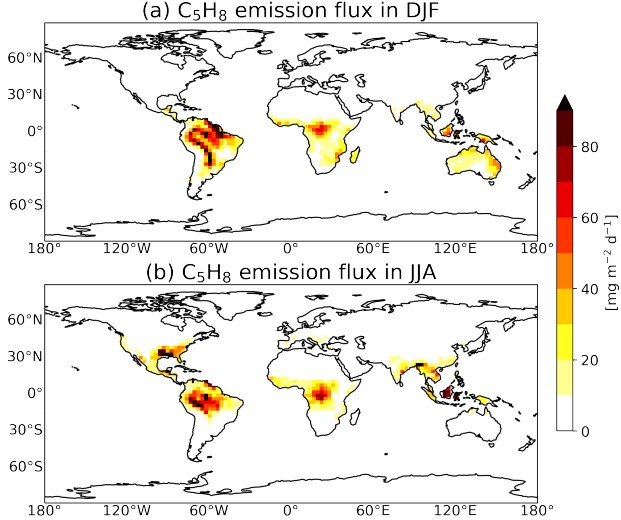

**Figure A3.** Mean isoprene mixing ratio near the surface in DJF (a) and mean isoprene mixing ratio near the surface in JJA (b) (EMACref)



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
