# Peer review of "The influence of weather-driven processes on tropospheric ozone"

_Atmospheric Chemistry and Physics, 2021_

## Author Comment (AC1)

**Answer to ACP referee 1**

August 4, 2021

**We thank Referee 1 for the detailed review of the manuscript and the constructive comments. For better readability, here we present again the comments together with our replies (in bold). This aims at explaining how the main referee's comments will be addressed in the revised manuscript. (We hope the Referee 1 to evaluate our answers before the closure of the open discussion in ACP(D)).**

This paper explores the impact of several changes to a global chemical transport model on tropospheric ozone, in particular the role of water vapor in chemical kinetics, dry deposition, and natural NOx, isoprene and HONO emissions. The authors have attempted to harmonize their study under the theme of 'weather-driven processes on tropospheric ozone', but their attempt fails. Not only are processes tied to meteorology in rather crude ways (which my main complaint is that the paper does not acknowledge), but the focus on annual and summer average global distributions is inconsistent with the theme of furthering understanding of the impact of weather-driven processes on ozone. The paper, including the background and motivation, lacks focus and direction, and often does not reflect current knowledge (including current understanding of processes examined). In particular, the papers referenced are often inappropriate, and the knowledge base cited is incomplete. In theory, this paper is much more suitable for Geoscientific Model Development, but needs substantial improvement, new analysis, and reframing, including clear articulation of what we are learning, before publication anywhere.

**Reply: The article reports on multiple sensitivity studies and their comparison with measurement data. We agree that the manuscript's focus needs to be better explained in the title. We propose to change the title to 'Reducing the tropospheric ozone bias in global models: the potential of processes modulated by weather'. The potential of each individual processes will also be more highlighted in the analysis section with a comparison of the averaged tropospheric ozone bias for the globe and specific regions.**
**We will go through the list of references and see where it can be complemented and/or updated. We kindly ask the Referee 1 to point to the references which he/she considers as inappropriate.**

Finally, we believe that the manuscript explores general implications for modelling tropospheric ozone and that model development is not its main focus. Instead, we combine atmospheric modelling, remote sensing data and ground-based measurements to investigate processes of global interest, which is clearly within the scope of ACP.

First, the authors alter the reaction kinetics related to HO2+NO and RO2+NO with respect to water vapor. Most of the impact is from HO2+NO. They discuss how this changes chemical production and loss of Ox as compared to other models. Then the authors discuss how this impacts tropospheric composition and dry deposition, but it is unclear what they find in their study that complements other work vs. what others find (that the authors are inferring happens in the model too). It seems like their findings are substantial in terms of tropospheric and surface ozone but honestly I have trouble following what they are doing and putting their findings in the context of uncertainty related to these changes to the reaction kinetics. The authors need to better describe motivate what they had before and why they made the change, and the associated uncertainty. The current motivation makes it seem like these changes are obvious/critical/completely certain, but then why has no one else made them?

Reply: The here used humidity-dependent HNO3 formation channel from the NO + HO2 reaction has been investigated in two global studies by Gottschaldt et al. (2013) and Righi et al. (2015) which give a recommendation to consider this kinetics for a realistic model representation of tropospheric ozone. They found a decrease of tropospheric ozone due to less efficient NO2 photolysis which reduces the tropospheric ozone bias. However, the JPL kinetic data evaluation panel (Publication 19-5) reports the water-dependence by Butkovskaya et al. (2009) for the purpose of model evaluation. No recommendation can be made by the panel until confirmation from other labs are published. The first but indirect confirmation has come from the characterization of a Chemical Amplifier instrument for measuring peroxy radicals (Duncianu et al. 2020). Box modelling of the chemistry inside the instrument supports the findings by Butkovskaya et al. (2009) although with a water effect half as strong. Therefore, for our study we have applied these modified kinetic data. In the manuscript we have also shown the potential importance of the HNO3-formation channel at the surface by comparing the model results to the TOAR measurement data.
Additionally, we have investigated the impact of potential large yields of alkyl nitrates in the RO2 + NO reactions in the presence of water. Like for the NO + HO2 reaction, complexes of water with peroxy radicals may significantly decrease the photochemical ozone formation. We had initially thought that it is potentially important for atmospheric modelling of tropospheric ozone but our sensitivity study shows that the impact is small.

The experimental kinetic data by Amedro et al. (2020) for the OH + NO2 reaction have been confidently included as it is from a well established kinetics lab at the Max Planck Institute for Chemistry. The established rate coefficient lies in between the recommendations of the two expert panels IUPAC and JPL. Their recommended kinetic data differs under dry conditions, especially in the cold UTLS by 50 %. Nevertheless, significant impacts of the water-effect in the OH + NO2 reaction on ozone production are expected only for very polluted and humid air masses. These conditions are seldom represented at typical spatial resolutions of 100-300 km by global models and have not been explored further in this study.
With this information, we will expand the relevant section of the revised manuscript in order to better motivate our work.

Second, the authors describe the impacts of a new dry deposition scheme (previously described by a paper this year in GMD with the same first author). The analysis on dry deposition focuses on surface ozone changes during morning vs. night and simply compares the total loss to dry dep to other models. My first complain is – what are we learning from this analysis? As is, this analysis does not merit publication. The difference changes to surface ozone during morning vs. night with dry deposition are intriguing, but the authors barely touch on it. My second complaint is that the authors pitch their representation of dry deposition as state of the science when it is not. From what I can tell, the authors' initial dry deposition scheme was not good, and there are only rather basic improvements to the new one.

**Reply: We did not intend to represent the dry deposition scheme used here as state of the art. However, we apply this commonly used approach due to it's simplicity and adaptability. The extension described in Emmerichs et al. (2021) considers more meteorological dependencies than the standard scheme, i.e. not only the light dependence for stomatal uptake but also temperature and vapour pressure deficit. These dependencies have been shown to improve the simulation of dry deposition fluxes (Wu et al., 2011, Val Martin et al., 2014). In addition, with the cuticular deposition the scheme includes a second uptake pathway by vegetation also active during night where the light-dependent stomatal deposition is low. A detailed analysis of how the modified dry deposition scheme affects the individual uptake pathways and the total dry deposition velocity of ozone (and other trace gases) is given in the GMD paper. Here, we show the potential of this modification to reduce the tropospheric ozone bias. By separating day- and nighttime values the large impact of the light-independent cuticular uptake is highlighted. The changes are presented for the Northern Hemisphere where most of the land is located and most of ozone measurements exists. This demonstrates the role of using a proper parametrization of a non-stomatal dry de-**

position pathway for tropospheric ozone.

**In the revised manuscript, we will better motivate the usage of the dry deposition scheme and clarify the outcome of the analysis with th information provided here. If Referee 1 have specific suggestion for extending the analysis we will consider them.**

Third, the authors describe a new drought dependence of isoprene emissions. The authors only really discuss how this changes annual total isoprene emission totals as compared to previous estimates. The authors barely discuss how this changes summer ozone, although there seem to be interesting changes in Fig. 6. As is, this analysis does not merit a publication.

**Reply: We investigate the impact of the drought stress factor for isoprene emissions on tropospheric ozone. This stress factor is part of the newest Model of Emissions of Gases and Aerosols from Nature (MEGAN3) which is of interest for the whole model community. Due to the plant stress isoprene levels are decreased in the major emission regions. Subsequently less peroxy radicals are produced, which lowers the NO-to-NO2 conversion via the reaction RO2 + NO by 13 % globally (-16 % in the PBL). Consequently, a significant decrease of surface O3 is seen. The change of the trace gases are in line with Jiang et al. (2018) who calculated the here used offline factor. To our knowledge, our study presents the first estimate of the change in the global isoprene emission and the related global ozone production due to the drought stress. We will make this clear in the revised manuscript.**

Fourth, the authors put a simple HONO soil emission parameterization into their model by scaling soil NO emissions by HONO/NO ratios from boreal field samples. As the authors note not many global models have soil HONO emission sources, so theoretically this is new. (The authors need to clarify if this is the first HONO soil emission parameterization in a global model, or one of the first; if the latter, please cite other work). However, I think the authors need to discuss their approach and the related uncertainties more. Just saying soil HONO and NO emissions are 'mechanistically similar' and applying boreal field sample values globally is not sufficient. (By 'boreal' – the authors mean northern mid latitude regions, right? Or do they mean northern hemisphere?). There also needs to be more substantive analysis of the impact of HONO. The authors discuss how HONO emissions impact the HONO/NOx ratio but do not describe what this ratio means, or why it should be one value over the other. (The reader is also completely relying on the author to summarize the model-observations comparison for HONO/NOx accurately – it is likely worth adding a figure comparing model vs. observed values.) The authors then discuss the impact on OH, NO2 and O3 extremely briefly, and with a lot of speculation. I would ask the authors to focus on what their plots are showing and what we are learning from their analysis, with a particular focus on addressing the goal of

the paper (which currently seems to be to address the impact of weather driven processes on ozone, but in my view needs to change).

**Reply: In order to address the importance of HONO emissions from soil on tropospheric Ox, HONO emissions from soil have been implemented here for the first time in a global model. The process has been proposed as missing model source for HONO by several studies. However, a parameterization for global models does not exist yet while it is available for regional models (doi:10.1002/2015JD024468). Hence, we decided to use the measured HONO/NO emission ratios by Oswald et al. (2013) determined for multiple soil samples from various ecosystems covering a range of different soil pH, organic matter and soil nutrient content (the term boreal doesn't belong here). Applied to the simulated NO soil emissions this estimates the emission characteristics of HONO at the respective vegetation types where for each of the four vegetation types (Table A1, 4 different vegetation categories: forest, woody savanna, grassland, crops) at least two measurements exist. In order to assign the measured data to the respective vegetation category we have used the International Geosphere Biosphere Programme (IGBP) land cover scheme (Belward, 1992). Obtaining characteristics for different vegetation types from observations is often used for the calculation of e.g. land-atmosphere exchange processes (e.g. Bousetta et al., 2011) which depends on the quality on the measured data and hence introduce some uncertainties. The measurements here used have an uncertainty of 1-4 ng m$^{-2}$ s$^{-1}$. Furthermore, the uncertainties of the NO soil emissions like the lack of a proper soil moisture dependence (Steinkamp et al., 2009) applies also for the emissions of HONO.**
**The analysis show a general agreement of the modelled range of HONO soil emissions with measured fluxes. In addition we give the global distribution of the HONO/NOx ratio to show a measure for the HONO mixing ratio considering the NOx conditions and also compare it with literature values. The enhanced HONO levels at the surface increases the global photolysis of HONO by 97.8 % in the troposphere (Table 3 showing the Ox budgets will be extended) compared to EMACref (more than double for the PBL). The enhanced formation of OH and NO, hence, increases the yield of the main Ox production by 4-7 % (double in the PBL) which leads to an enhanced mixing ratio of O3. These changes are highest on the SH where the NOx concentrations are generally lower. As long as the model does not contain the full heterogeneous HONO chemistry a direct comparison to measurements in not worth.**
**This information will be added to the relevant sections of the revised manuscript in order to better explain the used parametrisation and the outcome of this sensitivity study.**

Fifth, the authors put the Price and Rind lightning NOx scheme into their model and show how it changes NO2 (Figure 16c looks the same to me as Figure 16a though...so I'm not sure I see the improvement). They say it improves the model bias over tropical oceans and N Africa and the central US, but again I do not see this with their figures... They also show how it changes the tropospheric ozone bias compared to IASI (again, hard to see the improvement on Fig 12... I can see that there is a decrease in SOME parts of the tropical Pacific, but a worsening over others in 13c). Nonetheless, the authors' claim that "the usage of the commonly applied P+R scheme improves significantly the NO2 bias between EMAC and OMI as well as the trop O3 discrepancy to ISAI over the tropical Pacific Ocean" seems like an overstatement, and I'm not sure their current analysis on lightning NOx merits publication.

**Reply: The Price&Rind scheme is used for this study since it contains a more realistic representation of lightning activity over oceans, where we see an improvement potential in representing tropospheric ozone. We agree that the simulated change in Figure 16c is not well visible. The significant reduction we mention occur over the ocean where the NO2 levels are very low. Hence, we will also provide the relative bias while changing to a more intense colour scheme. Over the southern continents the Price&Rind scheme increases the lightning frequency which is in comparison to measurements an overestimation, also shown before in the literature. However, over land the NOx emission by lightning is not so important as over the ocean because of the background NOx levels. The range of the colour scale was chosen like this to allow a comparison of the changes among several modifications. In general, this sensitivity study shows the improvement potential for tropospheric ozone by using the Price&Rind scheme. Finally, we believe our results are relevant for the modelling community around the MESSy which participate to international model intercomparison projects like CCMI (Jöckel et al. 2016). In the global model configuration of MESSy (EMAC) the standard lightning NOx scheme is the one by Grewe, which we show to yield worse results over the tropical Pacific ocean. We will make the here mentioned changes in the revised manuscript and expand the relevant section with a more detailed discussion.**

Overall, while the updates to the model do not always seem to be state of the science, I understand that this is a global model with limited represented of processes like convective updrafts and land surface functioning and soil microbes, thus parameterizations need to be simple. I want to be clear that I'm not asking the authors to represent everything in a state of the science way, rather better articulate what the state of the science is, and justify their decisions in going the simple route. And of course the authors also need to convey what we are learning from their changes... not just document them. I also understand that the authors are limited because the surface ozone data for which

they use to evaluate their model is mostly for northern mid-latitudes, but many of the changes in ozone occur in the southern hemisphere and tropics, or high northern latitudes. This is an issue for all global modelers, and the authors need to find a better way of addressing this issue. For example, what are we learning from places where we do have observations that can lend confidence to elsewhere? What are other observational constraints that can be used to evaluate poorly sampled areas? There must be something better than one global map of the IASI tropospheric ozone columns and one global map of the OMI tropospheric NO2 columns.

**Reply: We know about O3 measurements at single stations which are not part of the TOAR data set such as the Amazonian rainforest (ATTO) or Bukit Atur (Malaysia) however the measurement data does not cover the here investigated period.**

Minor comments (I have many minor comments, only some of which I have taken the time to record here).

Line 3 – I don't know what 'the water complex' refers to

Line 5 – one of the most important reactions

Line 9 – I don't think it's true that most trop ozone loss to dry deposition occurs to vegetation, a lot of loss happens to snow and ice and water as recent papers such as Hardacre et al. 2015, Pound et al., 2020 (https://doi.org/10.5194/acp-20-4227-2020), Clifton et al., 2020 (https://doi.org/ 10.1029/2020JD032398) have suggested, despite relatively small deposition velocities over these surfaces

Line 27 – radiative forcing should only be discussed when referring to a change time period

Line 220-230 – will the authors illustrate the changes they are making as described here with equations? I find it very hard to figure out what they are doing vs. just discussing

Line 295-7 – are the authors suggesting here that O3 loss through reaction between O3+NO is underestimated here, and may cause the remainder of the high bias in O3?

Line 389 – can the authors give the uncertainty in a relative sense?

Line 390 – what is the IC/CG ratio?

Line 399 – who describes the Price and Rind scheme as robust?

Line 399 – it's not just a question of flash frequencies ... but also the amount of NOx per flash and how to distribute the NOx vertically

**Line 3: A water complex refers to a complex with one water molecule in the radical reactions.**
**Line 5: radical reactions important for O3 photochemistry.**
**Line 9: We agree and we will change that in the manuscript.**
**Line 27: We will remove this part.**
**Line: 220-230: We will add the reaction kinetics.**
**Line 295: Yes, this refers to the reaction of O3 with NO emitted by soil as proposed by Fares et al. (2013).**

**Line 389: The relative uncertainty of lightning emiisions of NOx is about 20 %.**
**Line 390: intra-cloud (IC)/cloud-to-ground (CG) flash ratio.**
**Line 399: Tost et al, 2007 describes the Price and Rind scheme as robust.**

Another minor issue is in their methods the authors often do not give the reader sufficient information. For example, can the authors elaborate on what 'weakly nudged' means? Also, discussing the model routines by their names in the model is not very helpful for the reader. Another example is how am I supposed to interpret Table A1 (what does number refer to, what is IGBP category and how is it used, what are the soil samples from? The title is unhelpful).

**Reply: The term 'weakly nudged' means that the dynamics (prognostic variables: logarithmic surface pressure, temperature, divergence and vorticity) are nudged by Newtonian relaxation to ECMWF reanalysis data (temperature, vorticity and divergence only between 4th model layer and 200hPa).**
**The explicit names of the MESSy submodels are mentioned in order to allow the reader to comprehend the used model set up.**
**The numbers in Table A1 refer to categories/vegetation types in the IGBP land cover scheme. The locations of the soil samples are shown in the supplementary material of Oswald et al. (2013) (`https://science.sciencemag.org/content/sci/suppl/2013/09/11/341.6151.1233.DC1/Oswald-SM.pdf`).**

**References**

Amedro, Damien, et al. "Kinetics of the OH+ NO 2 reaction: effect of water vapour and new parameterization for global modelling." Atmospheric Chemistry and Physics 20.5 (2020): 3091-3105.

Boussetta, Souhail, et al. "Impact of a satellite-derived leaf area index monthly climatology in a global numerical weather prediction model." International journal of remote sensing 34.9-10 (2013): 3520-3542.

Butkovskaya, Nadezhda, et al. "Water vapor effect on the HNO3 yield in the HO2+ NO reaction: experimental and theoretical evidence." The Journal of Physical Chemistry A 113.42 (2009): 11327-11342.

Duncianu, Marius, et al. "Characterization of a chemical amplifier for peroxy radical measurements in the atmosphere." Atmospheric Environment 222 (2020): 117106.

Emmerichs, Tamara, et al. "A revised dry deposition scheme for land–atmosphere exchange of trace gases in ECHAM/MESSy v2. 54." Geoscientific Model Development 14.1 (2021): 495-519.

Fares, Silvano, et al. "Testing of models of stomatal ozone fluxes with field measurements in a mixed Mediterranean forest." Atmospheric environment 67 (2013): 242-251.

Gottschaldt, K., et al. "Global sensitivity of aviation NO x effects to the HNO 3-forming channel of the HO 2+ NO reaction." Atmospheric Chemistry and Physics 13.6 (2013): 3003-3025.

Jiang, Xiaoyan, et al. "Isoprene emission response to drought and the impact on global atmospheric chemistry." Atmospheric Environment 183 (2018): 69-83.

Jöckel, Patrick, et al. "Earth system chemistry integrated modelling (ESCiMo) with the modular earth submodel system (MESSy) version 2.51." Geoscientific Model Development 9.3 (2016): 1153-1200.

Oswald, R., et al. "HONO emissions from soil bacteria as a major source of atmospheric reactive nitrogen." Science 341.6151 (2013): 1233-1235.

Steinkamp, J., et al. "Influence of modelled soil biogenic NO emissions on related trace gases and the atmospheric oxidizing efficiency." Atmospheric chemistry and physics 9.8 (2009): 2663-2677.

Tost, H., P. Jöckel, and J. Lelieveld. "Lightning and convection parameterisations–uncertainties in global modelling." Atmospheric Chemistry and Physics 7.17 (2007): 4553-4568.

Val Martin, M., C. L. Heald, and S. R. Arnold. "Coupling dry deposition to vegetation phenology in the Community Earth System Model: Implications for the simulation of surface O3." Geophysical Research Letters 41.8 (2014): 2988-2996.

Wu, Zhiyong, et al. "Evaluating the calculated dry deposition velocities of reactive nitrogen oxides and ozone from two community models over a temperate deciduous forest." Atmospheric Environment 45.16 (2011): 2663-2674.